# THE AFFINE DIVERGENCE: ALIGNING ACTIVATION UPDATES BEYOND NORMALISATION

**George Bird**
Department of Computer Science
University of Manchester
Manchester, UK
`george.bird@postgrad.manchester.ac.uk`

## ABSTRACT

A systematic mismatch exists between mathematically ideal and effective activation updates during gradient descent. As intended, parameters update in their direction of steepest descent. However, activations are argued to constitute a more directly impactful quantity to prioritise in optimisation, as they are closer to the loss in the computational graph and carry sample-dependent information through the network. Yet their propagated updates do not take the optimal steepest-descent step. These quantities exhibit non-ideal sample-wise scaling across affine, convolutional, and attention layers. Solutions to correct for this are trivial and, incidentally, derive normalisation from first principles despite motivational independence. Consequently, such considerations offer a fresh, conceptual reframe of normalisation's action, with auxiliary experiments bolstering this mechanistic interpretation. Moreover, this analysis makes clear a second possibility: a solution that is functionally distinct from modern normalisations, without scale invariance, yet remains empirically successful — an alternative to the affine map. This outperforms conventional normalisers across several tests. This generalises to convolution via a new functional form, "*PatchNorm*", a compositionally inseparable normaliser. Together, these provide an alternative mechanistic framework that both adds to and counters some of the discussion of normalisation. Further, it is argued that normalisers are better decomposed into activation-function-like maps with parameterised scaling. Overall, this constitutes a theoretically principled approach that yields new functions with empirical validation and raises questions about the affine + nonlinear approach.

## 1 INTRODUCTION

Deep learning models use two primary forms of variables to produce the desired outputs. These are the parameters tuned by optimisation, such as weights and biases, and the transient activations/representations, which depend on the specific input, are present in intermediate calculations and constitute the propagated information to the output.

The objective of deep learning is to make stepwise adjustments to such quantities to produce a meaningful output. This is undertaken by backpropagation and an optimiser, such as the gradient descent algorithm. Backpropagation computes the gradients of all intermediate quantities with respect to the loss; these gradients represent the direction of steepest descent that changes the loss the fastest with respect to that quantity. This calculation includes both activations and parameters, yet only the parameter gradients are used to update the model.

Activations cannot be directly descended because they are functions of the input, which determine their numerical value. Therefore, we update the parameters. These become tuned through time, which in turn influences activations. These, in turn, propagate to yield the desired result. Hence, in some respects, one can argue that parameters are updated as a practical proxy for desired changes in representations and the output.

In gradient descent, parameters are updated by subtracting the gradient of the loss with respect to the parameters, then scaling it by a learning rate scalar. This produces a slight change in the parameters in *their* direction of steepest descent with respect to the loss. For the batch of interest, in this 'parameter-picture', this is the optimal correction to reduce the loss the fastest.

However, the optimal steepest direction for adjusting activations to reduce the loss is also obtained from this calculation. The natural question arises: *Does the steepest descent for parameters correspond to those of representations when propagated?*

Counterintuitively, even in the simplest cases, this is not so.

This constitutes a fundamental structural misalignment in the update step in the *representation space* for affine maps[1]. A divergence arises between the theoretically ideal correction, analytically available from the representation's gradient, and the effective updates obtained by propagating the change in parameters into the representations. Yet, representations more directly influence the loss, and carry sample-dependent information that ultimately forward propagates to the overall network output. Motivating *why is the steepest descent for parameters prioritised whilst representations are allowed to diverge?* Moreover, can solutions be constructed such that they do coincide?

This paper explores the consequences of adapting computations until such an alignment occurs, forming the steepest direction in both parameters and the propagated representation corrections. Unexpectedly, the solutions to such a question naturally yield normalisation-like adaptations. Hence, normalisation and its placement are *derived* as a surprising consequence of enforcing alignment between parameter and representation updates, rather than as an a priori design assumption or empirically motivated.

Thus, provided is a fresh, novel perspective on the success of such implementations in addressing this theoretical 'affine divergence'. Empirical support for this alternative hypothesis is provided. Furthermore, auxiliary hypotheses are developed indicating that, if this affine divergence mechanism is valid, these new normalisers should, counterintuitively, negatively correlate performance with increases in batch size. This is also empirically validated, adding an independent hypothesis that supports this "affine divergence" as *a* mechanistic lens on the success of normalisation.

Moreover, a second solution is derived. It is neither normalisation-like nor scale-invariant, yet it is empirically successful and outperforms several forms of normalisation across tests. This peculiar and novel functional form's performance may seem classically surprising in the absence of the affine divergence explanation, and it provides a theoretical and empirical counterargument that partially undermines scale-invariance as a primary mechanistic cause of empirical success, motivating further investigation.

Extremising this divergence approach, by propagation of gradient corrections to the output layer, is conceptually aligned to the natural gradient descent approach (Amari, 1998; Amari et al., 2019; Martens, 2020). This approach considers the Riemannian geometry of the loss by instead employing the contravariant gradient $\nabla \tilde{\mathcal{L}} = G^{-1} \nabla \mathcal{L}$, which induces the Fisher information as the metric $G$. This effectively prioritises the steepest direction in the model's output function space rather than in parameter space, and is invariant under reparameterisation. However, despite conceptual alignment on the suboptimality of classical gradient descent's direction of steepest descent for the loss, this approach differs substantially from the methods presented in this paper in several key respects and is discussed further in *App.* D. Namely, differing variable prioritisations, entirely different approaches to solutions, and a large disparity in computational tractability.

This paper primarily raises considerations around the underappreciated 'ideal-effective' steepest-descent misalignment and its broader implications for model design. Encouraging consideration of which quantities should be given the highest priority in terms of update, and motivating new adaptations to resolve such divergences. This is suggested to be a widely applicable consideration across deep learning layers, extending further than this paper's explorative analysis[2].

---

[1] Primacy of parameters, despite prevalence, is not a neutral commitment either. It requires justification to claim the direction of parameters update *should* supersede that of representations, given its sample dependence.

[2] This was developed from theoretical discussion (Bird, 2025b) regarding the mechanistic explanation of isotropic activation functions' (Bird, 2025a) empirical results contrasted with their anisotropic counterparts.

The main text elucidates the divergence mathematics for affine layers and emphasises consideration not only of this divergence but also which updates bear the greatest responsibility for providing the corrections, assessing whether rebalancing this significantly alters learning. *App.* C extends these considerations into other architectures, including a novel convolutional 'PatchNorm' approach. Additionally, *App.* B argues to dissolve the distinction between what constitutes a normaliser and an activation function using an algebraic decomposition. This is used to foreground their geometrical interpretations rather than statistical ones, while encouraging this category unification and a decomposed treatment of normalisers more broadly for analysis. Overall, this work reimagines the standard construction of the affine/convolution + nonlinearity model by novel divergence-solving maps.

## 2 THEORETICAL BACKGROUND

This section provides a mathematical overview of the affine divergence, clarifying its origin and the discrepancy between the ideal and effective updates. Following this is a discussion of its implications, then the derivation of the various solutions to its convergence and their interpretations and properties.

### 2.1 DERIVING THE AFFINE DIVERGENCE

Propagation of corrections is required to assess whether the optimal update for parameters coincides with that for representations in affine layers. Considering the affine layer of *Eqn.* 1[3], assuming for now that the activation function is in a later step, e.g. $\vec{y} = \mathbf{f}(\vec{z})$.

$$\vec{z} = \mathbf{W}\vec{x} + \vec{b} \quad \Longleftrightarrow \quad z_i = W_{ij}x_j + b_i \tag{1}$$

Then a differentiation with respect to loss, $\mathcal{L}$, for each term is undertaken, with notation $\frac{\partial \mathcal{L}}{\partial \vec{z}} \equiv \vec{g}$.

$$\frac{\partial \mathcal{L}}{\partial z_n} = g_n \tag{2}$$

$$\frac{\partial \mathcal{L}}{\partial W_{nm}} = g_n x_m \tag{3}$$

$$\frac{\partial \mathcal{L}}{\partial x_n} = g_i W_{in} \tag{4}$$

$$\frac{\partial \mathcal{L}}{\partial b_n} = g_n \tag{5}$$

These represent the directions of steepest descent for *each* quantity with respect to the loss. Substituting these partial derivatives into the gradient descent update yields the familiar correction for *parameters* shown in *Eqns.* 6 and 7, with learning rate $\eta$.

$$W'_{ij} = W_{ij} - \eta\frac{\partial \mathcal{L}}{\partial W_{ij}} \quad \Rightarrow \quad W'_{ij} = W_{ij} - \eta g_i x_j \qquad [\Delta W_{ij} = -\eta g_i x_j] \tag{6}$$

$$b'_i = b_{ij} - \eta\frac{\partial \mathcal{L}}{\partial b_i} \quad \Rightarrow \quad b'_i = b_i - \eta g_i \qquad [\Delta b_i = -\eta g_i] \tag{7}$$

However, as argued, parameters act as a proxy to update the representations which are not directly amenable to update. Yet, their effective update is derivable by propagation of parameter corrections.

Computationally straightforward solutions are desired, necessitating several approximations (encouraged to be relaxed in future work). These are: single-step corrections and first-order in learning rate, and secondly, a single-layer approximation, i.e. the propagation begins at each affine layer's parameters and terminates at the affine layer's output activations. These are justified by first-order being already assumed in typical gradient descent and single-layer approximations, facilitating simple corrections that are easily implementable into networks with negligible computation[4] Hence, these

---

[3]Notationally, a non-tensorial form of Einstein Summation Convention is used throughout, and all terms should be evaluated at a particular point. This notation is suppressed for straightforward equations.

[4]Propagating through multiple non-linear layers makes such an approach analytically non-trivial and entails complicated solutions which may be intractable to determine/reasonably implement. Hence, this approximation avoids the activation functions, despite these considerations initially motivating this investigation (Bird, 2025b).

approximations are implemented to foreground simple, practical maps for networks that may have broader applicability. In this section, a single-sample assumption is made, but is relaxed in *App.* C.1.

Having established the need for such assumptions, the effective correction to representations can be analytically calculated, shown in *Eqn.* 8 and assume for the same sample/batch with $x'_j = x_j$. This can be compared against the mathematically ideal update, shown in *Eqn.* 2.

$$
\begin{aligned}
z'_i &= W'_{ij} x_j + b'_i \\
&= \left( W_{ij} - \eta g_i x_j \right) x_j + \left( b_i - \eta g_i \right) \\
&= z_i - \eta g_i \left( x_j x_j + 1 \right) \qquad\qquad \left[ \Delta z_i = -\eta g_i \left( x_j x_j + 1 \right) \right]
\end{aligned}
\tag{8}
$$

As $\eta \to 0$, this can be reorganised into *Eqn.* 9. This result is denoted $\Delta \mathcal{L} / \Delta z_i$, suggestive of a finite-difference 'effective gradient' interpretation as the argument is eventually to make this the desired analytical gradient (steepest descent); however, the notation should not be interpreted formally.

$$
\frac{\Delta \mathcal{L}}{\Delta z_i} = -\frac{z'_i - z_i}{\eta} = g_i \left( \|\vec{x}\|^2 + 1 \right)
\tag{9}
$$

This is inequivalent to the mathematically ideal update of *Eqn.* 2, diverging through the term $(\|\vec{x}\|^2 + 1)$, producing a sample-wise quadratic bias in the gradient step. This "affine divergence" is summarised by *Eqn.* 10 or the equality of *Eqn.* 11. This affine derivation can be trivially generalised for other maps, such as a qualitatively similar, patchwise divergence for convolution shown in *App.* C.2.

$$
\frac{\Delta \mathcal{L}}{\Delta z_i} \neq \frac{\partial \mathcal{L}}{\partial z_i}
\tag{10}
\qquad\qquad
\boxed{\frac{\Delta \mathcal{L}}{\Delta z_i} = \frac{\partial \mathcal{L}}{\partial z_i} \left( \|\vec{x}\|^2 + 1 \right)}
\tag{11}
$$

Despite gradient sample-dependence being familiar for parameters, uniquely this activation-propagated form is conceptually central to this work; moreover, it is reframed as pathological.

## 2.2 Implications of the Affine Divergence

From the prior subsection, it is evident that the representations do not take their optimal step, yet are arguably a more direct quantity of interest, due to carrying sample-dependent information. In the absence of parameter-decay regularisations, they are also more closely associated with the loss than any parameter they depend upon, since they appear subsequent to them in the computation graph.

Important considerations this manuscript aims to develop are: ***Implications:*** *any detrimentality of the divergence?* ***Priority:*** *whether the parameters should take precedence at the expense of representations.* ***Corrections/Mitigations:*** *for the divergence, especially concurrently for parameters and representations.* ***Consequences:*** *both theoretical and empirical outcomes of such corrections.*

Implications of the divergence are primarily geometric. The sample's squared-magnitude acts as an unintentional weighting in the gradient, introducing a geometric inconsistency in the representation's effective step: large-magnitude samples produce disproportionately sized updates stepwise, or angularly distorted, over-weighted updates batchwise. Deflection occurs away from the ideal update trajectory or ideal steepest descent, respectively. This infers a conceptual suboptimality for learning.

The former can be interpreted as each sample's update step having updates of various effective learning rates, $\eta_{\text{eff.}} = \eta(\|\vec{x}\|^2 + 1)$. The latter, over a batch, is seen as a few large samples dominating the direction of the resultant representational gradient step relative to others, thereby skewing the direction of the update in their favour. Both scenarios introduce a foundational bias in affine optimisation.

Already apparent is that normalisation may reduce such magnitudes by altering activation distributions; however, it is hypothesised that such a relation would be better framed in reverse: *Perhaps the success of normalisation is due to approximately correcting the representational update*. This consideration will coincide with affine divergence solutions, which incidentally appear normalisation-like.

This hypothesis is intriguing, prior BatchNorm (Ioffe & Szegedy, 2015) offers only a partial mitigation, typically reducing $\text{Var}(\|\vec{x}\|^2[+1])$ by confining $\vec{x}$'s distribution; but, not identically cancelling this

term. Averaging gradients over batches may similarly help, but introduces nuance (*App.* C.1). Despite mitigation, these sample-wise fluctuations persist. LayerNorm (Ba et al., 2016) and RMSNorm (Zhang & Sennrich, 2019) operate differently; cancelling the term samplewise, yet introducing caveats: rebalancing the weight-bias responsibility in the propagated corrections, altering the effective learning rate, and losing geometric degrees-of-freedom (DOF) in backward and forward passes (*App.* B).

Hence, this affine divergence term may contribute to a fundamental mode in the success of current normalisations. Using purpose-built solutions also opens the possibility for further improvement, if this is a mechanistic, causal mode. This is undertaken in the following section: analytically deriving several straightforward and novel solutions to the divergence.

A number of these are mathematically distinct from current implementations and, therefore, are of particular interest since they do not exhibit features typically attributed to normalisation's success. Thus, a comparative ablation trial against other implementations provides insight into the validity of the stated hypothesis, if they equal or improve performance. This indicates whether the correction of the affine divergence or instead other, more classical, factors primarily underlie empirical performance.

## 2.3 Deriving the Affine Corrections

The objective of this work is to explore the consequence when *Eqn.* 12 holds true.

$$\frac{\Delta \mathcal{L}}{\Delta \vec{z}} = \frac{\partial \mathcal{L}}{\partial \vec{z}} \tag{12}$$

Infinitely many approaches exist to achieve such an equality; however, four *forms* will be primarily explored, but should be considered a non-exhaustive set of solutions.

Solutions that produce exactly ideal parameters *and* representational gradient steps by altering the affine mapping will be termed "structural" corrections. These solutions affect both forward propagation and backpropagation non-trivially, including changing the parameter updates. These can be summarised by the functional form shown in *Eqn.* 13, with $\alpha_{ij} \equiv \alpha_{ij}(\vec{x})$ and $\beta_{ij} \equiv \beta_{ij}(\vec{x})$.

$$y_i = \alpha_{ij} W_{ij} x_j + \beta_{ij} b_i \tag{13}$$

Different choices produce different families of solutions. Example choices are various contractions and equalities, e.g. $\alpha_{ij} = \alpha_i$, $\alpha_{ij} = \alpha$ or $\alpha_{ij} = \beta_{ij}$ etc. Two solutions primarily emerge: affine-like (*Eqn.* 15) and normalisation-like (*Eqn.* 14) offering further subdivision[5][6].

$$z_i = W_{ij}\left(\frac{x_j}{s}\right) + b_i \tag{14} \qquad\qquad z_i = \frac{W_{ij} x_j + b_i}{s} \tag{15}$$

To then determine the effective representational corrections, the relevant updates must be propagated from parameters into representations, yielding *Eqn.* 16 (norm-like) and *Eqn.* 17 (affine-like).

---

[5]Reweighting of these by $\alpha \in \mathbb{R}$, $\beta \in (-1, 1)$ provide an infinite family of structural maps. Different balances between weights and bias contributions change the updates and can be used to deduce implications. For $z_i = \alpha W_{ij}\left(\frac{x_j}{s}\right) + \beta b_i$, $s = \sqrt{\frac{\|\vec{x}\|^2 \alpha^2}{1 - \beta^2}}$ whilst $z_i = \frac{\alpha W_{ij} x_j + \beta b_i}{s}$ yields $s = \sqrt{\alpha^2 \|\vec{x}\|^2 + \beta^2}$.

[6]Equivalently for affine-like solutions, the bias can also be stacked into weights $\tilde{\mathbf{W}} = [\mathbf{W}, \vec{b}]$, with corresponding $\tilde{x} = [\vec{x}, 1]$, to notationally unify approaches. This is not undertaken for clarity, but results in an identical implementation only differing through notation.

$$
\begin{aligned}
z_i' &= W_{ij}' \left( \frac{x_j}{s} \right) + b_i' \\
&= \left( W_{ij} - \eta \frac{g_i x_j}{s} \right) x_j + (b_i - \eta g_i) \\
&= z_i - \eta g_i \left( \frac{x_j x_j}{s^2} + 1 \right) \Rightarrow s = \|\vec{x}\| \\
&= z_i - 2\eta g_i \\
&= z_i - \eta' g_i
\end{aligned}
\qquad (16)
$$

$$
\begin{aligned}
z_i' &= \frac{W_{ij}' x_j + b_i'}{s} \\
&= \frac{1}{s} \left( \left( W_{ij} - \eta \frac{g_i x_j}{s} \right) x_j + \left( b_i - \eta \frac{g_i}{s} \right) \right) \\
&= z_i - \eta g_i \left( \frac{x_j x_j + 1}{s^2} \right) \Rightarrow s = \sqrt{\|\vec{x}\|^2 + 1} \\
&= z_i - \eta g_i
\end{aligned}
$$

$$(17)$$

Notable is the effective doubling of the learning rate, which should be compensated with a halving of the hyperparameter to ensure comparability. Both $\eta$ and $\eta'$ will be used in experiments to demonstrate this. This half-factor can also be absorbed into $\alpha_{ij}$ and $\beta_{ij}$ if desired, using the generalised formalism.

Substituting $s$ derives the two structural corrections: *Eqn.* 18 (norm-like) and *Eqn.* 19 (affine-like). These solutions offer maps which identically cancel the affine divergence, ensuring both parameters and representations now take the mathematically ideal update step in the direction of steepest descent.

$$
\boxed{\vec{z} = \mathbf{W} \left( \frac{\vec{x}}{\|\vec{x}\|} \right) + \vec{b} \qquad \left( = \mathbf{W}\hat{x} + \vec{b} \right)} \qquad (18)
$$

$$
\boxed{\vec{z} = \frac{\mathbf{W}\vec{x} + \vec{b}}{\sqrt{\|\vec{x}\|^2 + 1}}} \qquad (19)
$$

The norm-like solution is effectively a classical (parameterless) $L_2$-normalisation, similar to (parameterless) RMSNorm without a $\sqrt{n}$ width factor. This implies that the affine divergence can derive a normaliser from first principles. Yet, the second, affine-like, solution also remedies the affine divergence, despite *not* being a normaliser but instead a modified affine map.

Empirical success of this latter non-normaliser form would situate the divergence as *an* underlying, mechanistic explanation for the success of such approaches as this novel functional form is wholly distinct from a classical normaliser. Thus, joint success would be attributable to the resolution of the divergence. The affine-like map has several further desirable qualities discussed in the next section.

The second entirely distinct approach (discussed further in *App.* D), under which natural gradients could be classified, is to transform the gradient updates only, similar to introducing an amended learning rate. These will be termed "gradient-only" corrections. They do affect the forward propagation of activations in subsequent steps, but do not fundamentally change the form of maps used.

Taken together, this divides solutions into "structural" versus "gradient-only". Importantly, each of which can be further subdivided in terms of weight-bias proportionality. This is argued to indicate the 'responsibility' of each parameter in the relative share of the propagated update. In effect, the ratio $\Delta\mathbf{W}$-to-$\Delta\vec{b}$ constituting $\Delta\vec{z}$ in the equation $\Delta\vec{z} = (\Delta\mathbf{W})\vec{x} + (\Delta\vec{b})$. If this is dominated by the bias, then the correction is primarily translationally acting as a global sample-independent offset. The relative responsibilities may then be tuned. The affine-like map preserves the proportions of the original affine map due to equal-term scaling. The norm-like update scales only the weights, resulting in a disproportionate effect. A similar subdivision applies to gradient-only approaches.

These considerations extend to the interesting $\sqrt{n}$ factor in RMSNorm, which serves as an up-scaling of the weight-responsibility relative to the $L_2$-Norm, *despite equivalence under reparameterisation*. These scaled proportions would shift more onus to the weights in the representational correction — further emphasising the weights as where the representational correction is largely realised. Rebalancing remains possible whilst resolving the divergence, but its potential impact must be carefully considered. Overall, this perspective has the potential to explain the empirical necessity of $\sqrt{n}$ for RMSNorm, and can be explored by altering the proportion in the divergence solutions.

## 2.4 PROPERTIES OF THE AFFINE STRUCTURAL CORRECTIONS

All of these corrections are consequential, particularly the structural approaches which have *direct* implications on both the forward and backward pass. In contrast, the gradient-only approach has direct implications only for the backward pass, from which an indirect effect on the forward pass

emerges through the parameters. Due to space restrictions, this section briefly outlines properties of the corrections. These are developed in greater detail in *App.* A, including associated mathematics.

Forward pass implications differ. The affine-like map acts as a soft bound on the representations, which preserves all initial DOF. This contrasts with the norm-like map, which irreversibly projects out the radial degree of freedom by a map onto a spherical shell. This does enforce scale-invariance, but at the cost of information loss, constituting the map $\mathbb{R}^n \to S^{n-1} \hookrightarrow \mathbb{R}^n$ a projection to the unit-hypersphere manifold, then a re-embedding into $\mathbb{R}^n$ — obfuscating the DOF loss. This information bottleneck requires a strong justification for implementation, and the magnitude cannot be assumed a priori to be redundant or meaningless; yet, radial modulation in the affine map may learn to suppress this degree of freedom if beneficial. Moreover, the loss in representational-DOF cannot be compensated for with new parameters, as they are in-substitutable quantities. Overall, both maps act as nontrivial nonlinear mappings, a decomposition argued in *App.* B, yet affine-like uniquely preserves all DOF compared to all previous normalisations.

Norm-like solutions also feature undesirable singular behaviour as $\|\vec{x}\| \to 0$, a genuine singularity unlike isotropic activation function's (Bird, 2025a;b) smooth coordinate form. The theoretically ad-hoc $\epsilon$-positivity trick can help mitigate this, but it does not resolve the latter exploding gradients argument and quietly reintroduces the divergence with additional $\epsilon$-distortion.

Thus, the comparable success of affine-like cannot be attributed to the scale-invariance of the forward pass, and the backward pass (also used to motivate RMSNorm's form). This backwards-pass departure can be seen in comparison of *Eqns.* 20 and 21, for norm-like and affine-like, respectively.

$$\frac{\partial \mathcal{L}}{\partial \mathbf{W}} = \vec{g}\hat{x}^T \qquad (20) \qquad \frac{\partial \mathcal{L}}{\partial \mathbf{W}} = \sqrt{\frac{\|\vec{x}\|^2}{\|\vec{x}\|^2 + 1}}\, \vec{g}\hat{x}^T \qquad (21)$$

Where *Eqn.* 20 features parameter-gradient scale-invariance, *Eqn.* 21 instead smoothly limits to the same $\vec{g}\hat{x}^T$ as $\|\vec{x}\| \to \infty$ but is non-singular as $\|\vec{x}\| \to 0$ without need for an $\epsilon$-term violating the solution. Conceptually, the magnitude modulation for *Eqn.* 21 may be desirable since near-zero representations do not cause significant parameter change, unlike norm-like. Practically, they become closely comparable when $1 \ll \|\vec{x}\|$.

Heuristically, this may also be desirable, considering a magnitude (strength) and direction (semantic) hypothesis for continuous representations. Crucially, gradients with respect to $\vec{x}$ indicate norm-like's liability to explode yet stable behaviour for affine-like due to a greater-than-one value of its denominator, shown mathematically in *App.* A. This suggests a theoretical preference towards affine-like. Additionally, reintroducing parameterised scalings may partially mitigate vanishing gradients, offering an explanation for their co-occurring empirical success.

The underappreciated proportionality change introduced by norms must also be explored. Determining implications caused by shifting the onus of which parameter more significantly updates. Norm-like alters this for the bias, making it more 'responsible' for the impact on representations. This may be undesirable as bias constitutes a simple, global translation, rather than a position-dependent map. As stated, this possibly explains the empirical necessity of $\sqrt{n}$ rescaling (Zhang & Sennrich, 2019).

Together, these analyses suggest a forward- and backwards-mechanistic advantage for the affine-like map over traditional functional forms and indicate that any success is not attributable to conventional explanatory modes, given their absence. This positions the novel affine-like solution as an interesting and discriminative map with respect to several classical explanations, motivating the following ablation tests.

## 3 RESULTS AND DISCUSSION

Provided are a range of ablation trials for parameterless normalisers and the affine-like map to fairly determine comparative performances. Further experimental details are given in *App.* E. Fully connected CIFAR10 (Krizhevsky et al., 2009) networks are used as a straightforward model to establish results where single-layer approximations remain within a valid regime, without many prior layers compounding into substantial propagated corrections. Fully connected models are necessitated by the nature of the *affine* correction, with various layer widths, depths and activation functions tested.

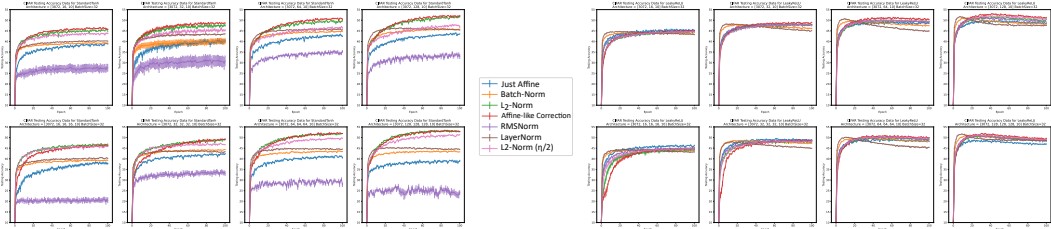

Figure 1: Displays CIFAR10 classification accuracy for fully connected networks, trained using batches. Individual titles indicate activation function, width and depth. Left-sided columns indicate Tanh, whilst right Leaky-ReLU. Especially for standard-tanh affine-like correction, it outperforms all other normalisers except for 3-layer 16-width, where norm-like structural correction marginally outperforms. Tanh results show structural corrections, affine-like or norm-like ($\eta$ & $\eta/2$), consistently and significantly perform better than all other non-divergence-corrective normalisers by a wide margin, even more so for deeper networks. RMSNorm underperforms, likely due to the parameterless form. Leaky-ReLU is more nuanced: early learning is faster across all normalisers, with a subsequent drop in performance for BatchNorm and LayerNorm only, before stagnation — sometimes observed in other normalisers on longer timescales. Nevertheless, the affine-like correction typically performs optimally, particularly at larger widths and depths, exhibiting a more distinct separation from all other normalisers. For very narrow networks (n=16), no normaliser performs well, likely because the norms' representational DOF reduction is a larger fraction for small width. The clearest performance separation is observed for the single hidden-layer n=128 result; the other Leaky-ReLU plots are more overlapping. Consistently, structural corrections significantly outperform alternatives.

*Fig.* 1 demonstrates the performance of various normalisers and structural corrections for both Standard Tanh and Leaky-ReLU. Width and depth vary as indicated by individual plot titles; each result reports the mean over 5 repeats with standard error, totalling 560 networks and 112 distinct model permutations.

These results hold across more widths, as shown in *Fig.* 2's more detailed trend, and results of *App.* C.

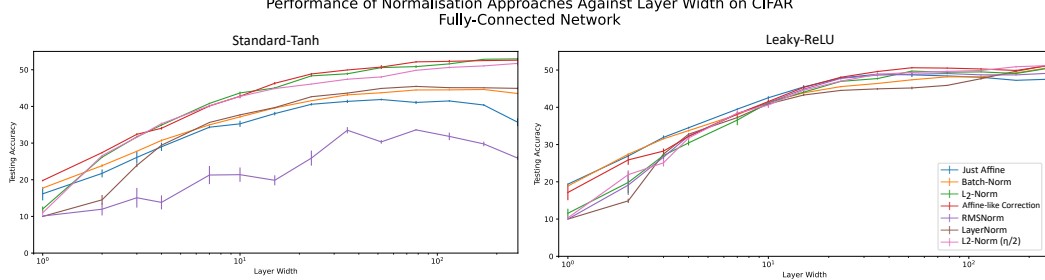

Figure 2: Plotted is performance against log-layer-width for Tanh (left) and Leaky-ReLU (right), with widths ranging from 1 through 256. Training is otherwise identical to *Fig.* 1. Tanh shows better performance separation, with the affine-like and norm-like structural solutions demonstrating significantly higher performance that grows with width. The affine-like solution also performs notably better, even at a width of 1 neuron. Both no-normaliser and RMSNorm peak at $n = 52$ and then decline in performance. For Leaky-ReLU, the differences are slighter, but Affine-like shows better performance at larger widths, while no normaliser outperforms at shallower widths.

Taken together, the trends indicate that solutions that address the 'Affine Divergence' perform strongly against all other normalisers. Notably, the affine-like formulation tends to be optimal, as suggested by its more favourable characteristics, despite not being a classical normaliser or featuring scale invariance. The performance distinction becomes more evident in wider and deeper networks. These results support a more geometric than a statistical interpretation of the normaliser's action. Overall, these results strongly suggest that the affine divergence provides a novel mechanism by which normalisers may operate and offer some counter-evidence to the scale-invariance theory. This

encourages further analysis of the ideal-effective misalignment hypothesis and broader evaluation of generalised structural corrections in architectures to determine any practical performance benefits.

## 4    CONCLUSION

A systematic mismatch between the ideal and effective updates taken by representations during optimisation is explored. A theoretical divergence is derived across fully-connected, convolutional, residual, and attention layers, suggesting the existence of novel approaches to correct for the misalignment.

Two structural correction families arose for the affine layer. One solution resembles the form of a normaliser, despite being motivationally independent. This offers an a priori derivational justification for the use of normalisers in addition to post-hoc explanatory modes and empirics. Hence, an additional mode for the success of normalisation is developed and empirically substantiated. The development chronology of normalisers appears to progressively mitigate this divergence unintentionally.

The other solution family differs markedly. The affine-like correction does not appear as a typical normaliser or display its characteristic properties, and therefore, its efficacy is unsupported by conventional theory. Nonetheless, derived as a distinct functional-form solution to the divergence, it tends to empirically exceed the performance of prior normalisers across a range of affine architectures. This supports the *existence* of the divergence as a valid, theoretical mechanism, although it is not to be mistaken as arguing for widespread practicality of solutions, until it is mechanistically generalised.

Secondary results of *App.* C.1, reinforce conclusions by predicting a counterintuitive, negative correlation between batch size and performance due to cross-sample interferences producing further ideal-effective misalignment. This auxiliary prediction is validated, strengthening support for the misalignment theory as a theoretical explanation for normaliser success rather than being incidental.

Results may also explain isotropic functions' success by providing better effective-ideal alignment than anisotropic forms (Bird, 2025b); additionally, the affine-like solution appears to be functionally similar to isotropic-tanh (Bird, 2025a) — supporting the unification argued in *App.* B, encouraging normaliser decomposition parameterised scalings and maps equating to activation functions. This situates modern normalisers as geometric operators above statistical ones in analyses, such as reweighting LayerNorm's mean to an extreme one-hot weighted statistic with no geometric consequence.

Appendix C develops the theory for various other maps, demonstrating the limits of the approximations that underpin the approach. Importantly, a novel form of convolution, with in-built normalisation, termed "**PatchNorm**", which is derived and tested. This functional form *family* represents a unique non-compositional approach to normalisation that is inseparable from convolution. Both residual and attention divergences are also outlined, with speculation that attention's lack of normalisation may offer weak, but circumstantial evidence, for the theory overall. Future work could investigate the relation to the optimiser choice. The ramifications of rescaling by running statistics on *representation* updates could be further studied, such as ADAM's element-wise rescaling (Kingma & Ba, 2017). Rescalings by statistical estimates do result in some deflection to the *parameter's* steepest-descent path per batch, but implications for *representation* deflection remain unclear, especially cumulatively. Nevertheless, results using ADAM show that the proposed considerations still yield improvement.

Overall, the ideal-effective misalignment theory independently provides a further mechanistic explanation of normalisers by correction of the 'affine divergence'. The theory also predicts further phenomena that are empirically validated. The normalisation-like map is derived, rather than assumed, using a priori theory, and a secondary affine-like map is evident as a further solution. The affine-like solution performs similarly or better, which cannot be explained by scale-invariance, as it does not feature it. Overall, this suggests that the ideal-effective first-order misalignment theory may warrant further investigation of its scope and empirical practicality, and that a philosophical shift in which corrections are prioritised may be a productive reconsideration more broadly.

ACKNOWLEDGMENTS

This work was supported by the Engineering and Physical Sciences Research Council (EPSRC), Grant number: $[EP/W524347/1]$.

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

# A    PROPERTIES OF THE AFFINE STRUCTURAL CORRECTIONS CONTINUED

This section further discusses properties of the structural corrections with greater and mathematical details. Discussion of gradient-only solutions is provided in *App.* D, with explanation as to why these solutions are sidelined in this work.

## A.1    FORWARD-PASS PROPERTIES

Previously discussed was the comparison of norm-like to affine-like structural corrections on the forward pass. This centred around the irreversible loss in a representational degree-of-freedom enacted by the norm-like solution's nonlinear map — analogous to other normalisations (Ioffe & Szegedy, 2015; Ba et al., 2016; Zhang & Sennrich, 2019) projections discussed in *App.* B and depicted in *Fig.* 3.



Figure 3: Displayed is the effect of various normalisers acting on an initial random sample of a thousand input points (green) drawn from a standard multivariate normal distribution in $\mathbb{R}^2$. These initial input points are consistently depicted in green with the action of the normaliser in a contrasting colour. Leftmost demonstrates BatchNorm's output as red mapped points, with statistics computed over all 1000 input points. The output distribution differs little from the input as the distribution was already standard-normal and statistics were well approximated over 1000 samples. The next column depicts LayerNorm, which can be seen to collapse the entire distribution into two single blue clusters, since the map follows $\mathbb{R}^n \rightarrow S^{n-2} \hookrightarrow \mathbb{R}^2$. For $n = 2$, this results in $S^0$, which preserves only sign information, yielding two isolated clusters. Typically, this would be a hypersphere orthogonal to $\vec{1}$ in higher dimensions[7]. Centre is $L_2$-norm, which projects the distribution into an $S^{n-1}$ purple hypersphere. Similar is the next right, with RMSNorm projecting to a $\sqrt{n}$ scaled hypersphere. Finally, a depiction of Affine-Like, for $\mathbf{W} = \mathrm{I}_{n \times n}$ and $\vec{b} = 0$, shows how the distribution fills the volume of the hypersphere without projecting out the radial representation degree of freedom.

Geometrically, this non-injectively projects activations onto a hypersphere before embedding back into the real space: $\mathbb{R}^n \rightarrow S^{n-1} \hookrightarrow \mathbb{R}^n$. This projection is how the scale-invariance manifests, as scaling does not alter the projection. Embedding back to the real space obfuscates such an information bottleneck, making the action seem geometrically innocuous, yet removing the radial information. Moreover, no increase in parameter degrees of freedom can compensate for a representational degree of freedom reduction, as they are fundamentally incommensurable quantities — they are not able to learn to replace sample-dependent information unless redundancy is guaranteed. Consequently, there is a distinct change in the represented information before and after the map.

A priori, it should not be assumed that such a degree of freedom is redundant or meaningless, particularly prior to training, nor that radial information is manifestly unimportant. To discard such information through projection requires justification or is otherwise undesirable. Treatment of this as a meaningless statistic would also require support that the radial information is inherently less informative.

---

[7]Fascinatingly, LayerNorm's standard mean subtraction, $\vec{x} - (\vec{x} \cdot \hat{1})\hat{1}$, can be reweighted to **any** weighted mean represented by $\vec{x} - (\vec{x} \cdot \hat{n})\hat{n}$. This has geometrically no effect in the absence of an external distinguished direction, except for reorienting the hypersphere to be orthogonal to $\hat{n}$ — acting as an unappreciated gauge freedom. Crucially, this extends to any one-hot weighting, $\hat{n} = \hat{e}_i$, undermining the mean-statistic argument used to often motivate LayerNorm. This one-hot weighting statistic is unaffected by layer width and becomes difficult to justify as an inherently redundant, nuisance statistic. If the one-hot *active* transformation is seen as troubling, it is equivalent to a *passive* basis transform for the unweighted mean. This supports the transition from statistical interpretation to a geometrical one, aligning with the activation function decomposition suggested in *App.* B.

This differs substantially from the affine-like map, which operates more like a nonlinear soft bound rather than an information bottleneck. The map's norm is displayed in *Eqn.* 22.

$$B\left(\vec{x}; \mathbf{W}, \vec{b}\right) = \|\vec{z}\|^2 = \frac{\vec{x}^T \mathbf{W}^T \mathbf{W} \vec{x} + 2\vec{b}^T \mathbf{W} \vec{x} + \left\|\vec{b}\right\|^2}{\|\vec{x}\|^2 + 1} \tag{22}$$

It can be non-invertible, due to *learnable* non-injective folding, but it remains monotonic (*Eqn.* 23)along each origin-centred ray for zero-bias nor explicitly remove a degree-of-freedom — only learned folding can suppress information redundancy.

$$\frac{\partial}{\partial \alpha} B\left(\alpha \hat{x}; \mathbf{W}, \vec{0}\right) = \frac{\partial}{\partial \alpha} \frac{\alpha^2 \|\mathbf{W}\hat{x}\|^2}{\alpha^2 + 1} = \|\mathbf{W}\hat{x}\|^2 \frac{2\alpha}{(\alpha^2 + 1)^2} > 0 \quad : \quad \alpha > 0 \tag{23}$$

With this nonlinearity, it may act as an activation function. Thus, the sequential affine + nonlinear sequential composition, *may* be unnecessary when using the affine-like map. However, the norm-like solution does represent a decomposable and therefore separate map from the typical affine layer. The decomposition is as if the affine layer precomposed with a normaliser, hence the association[8]. Generally, both structural solutions act nonlinearly on representations when replacing the affine layer, but can also be composed with further nonlinearities if desired.

Overall, suppression/preservation of such degrees of freedom, over initially unlearned absolute removal, is one arguably favourable characteristic of the affine-like solution over other normalisations.

### A.2 COMPARISON TO ISOTROPIC ACTIVATION FUNCTIONS

Also interesting is that the structural corrections to the affine divergence could both be considered isotropic-like in their effect, e.g. $\mathbf{f}\left(\|\vec{x}\|\right)\hat{x}$ — particularly in the absence of bias and identity-weights. Although the norm-like solution is not smooth and features a real singularity, unlike the isotropic activation function definitions. However, the natural emergence of isotropic-like functions in the corrections is intriguing, particularly since the affine-like correction on the vector $\vec{x}$ is analytically reminiscent of Isotropic-tanh (Bird, 2025a), especially given the aforementioned parameterisations. This algebraic comparison is shown in *Eqns.* 24 and 25, for the affine-like map and isotropic-tanh (post-composed with an affine layer), respectively, and without the bias contributions.

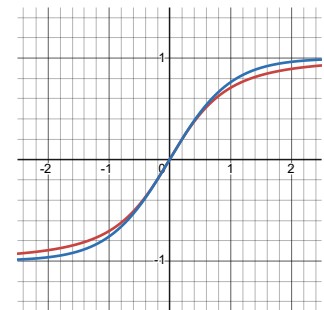

$$\mathbf{W} \underbrace{\sqrt{\frac{\|x\|^2}{\|x\|^2 + 1}}\, \hat{x}}_{\mathbf{f}(\|\vec{x}\|)\hat{x}} + \cdots \tag{24}$$

$$\mathbf{W} \underbrace{\tanh\left(\|x\|\right)\hat{x}}_{\mathbf{f}(\|\vec{x}\|)\hat{x}} + \cdots \tag{25}$$

Figure 4: Red denotes *Eqn.* 24 and Blue *Eqn.* 25, both in 1D: $\mathbf{f} : \mathbb{R} \to \mathbb{R}$ with $\mathbf{W} = 1$ and $\vec{b} = 0$.

Isotropic functions initially motivated the ideal-effective misalignment analysis (Bird, 2025b), and the coincidental alignment could partially explain some of the success behind isotropic-tanh. However, it does not explain the effectiveness of isotropic functions more broadly. More generally, this novel ideal-effective alignment theory remains pertinent to isotropy as previously suggested (Bird, 2025b). Overall, this serves to draw further geometric comparability between normalisers and activation functions, supporting the dissolution of the standard category distinction.

---

[8]Here 'normaliser' is used non-standardly. In effect, the parameterised scaling and the overall map are treated as two separate, decomposable contributions for study. The latter becomes essentially indistinguishable from the notion of an activation function, blurring their distinction.

A.3 BACKWARD-PASS PROPERTIES

It is essential to also characterise the effect that both the normalisation-like and affine-like structural corrections have on the backward pass. The standard affine layer backward pass with respect to input is shown in *Eqn.* 26, normalisation-like in *Eqn.* 27 and affine-like in *Eqn.* 28. Notationally, the affine layer output is given as $y_i = W_{ij}x_j + b_i$, $g_i$ is as denoted before and $s$ is used exclusively for the affine-like definition $s = \sqrt{x_j x_j + 1}$.

$$\frac{\partial \mathcal{L}}{\partial x_n} = g_i W_{in} \tag{26}$$

$$\frac{\partial \mathcal{L}}{\partial x_n} = g_i \left( \frac{W_{in}}{\|\vec{x}\|} - \frac{W_{ij}x_j x_n}{\|\vec{x}\|^3} \right) \tag{27}$$

$$\frac{\partial \mathcal{L}}{\partial x_n} = g_i \left( \frac{W_{in}}{s} - \frac{y_i x_n}{s^3} \right) \tag{28}$$

These equations demonstrate that for structural corrections, the backwards-propagating gradients acquire a dependence on the input $\vec{x}$. Hence, these modified maps carry substantial implications for gradients in deeper models, by reintroducing vanishing and exploding gradients through dependencies on $\|\vec{x}\|$ if not bounded.

Crucially, however, the affine-like map does exhibit bounding. The $\vec{x}$-dependent exploding possibility appears absent in *Eqn.* 28, as the denominator, $s$, must be larger than 1 by definition. This appears favourable over the norm-like behaviour, as the affine-like map is gradient stable.

Considering also the weight and bias derivatives, it has been shown that affine-like maps do not feature $\|\vec{x}\|$ invariance, unlike norm-like maps. However, the propagated representational corrections do, by definition. Therefore, it may also be explored whether it is this representational corrections' invariance to the magnitude (and direction for single samples) that could mechanistically additionally explain the success.

Determining whether the propagated update being unperturbed geometrically by $\vec{x}$ is causal, with the invariance forward-pass and weight updates for norm-like being coincidental to this, could be an interesting research direction and is encouraged by the performance of the affine-like map.

This could be significant, since gradient considerations partially motivated RMSNorm (Zhang & Sennrich, 2019), yet affine-like maps performed successfully and cannot be attributed to these properties. Therefore, examining this disparity might provide a fuller understanding of such implications, helping to pinpoint and generalise these properties as motivations for new maps.

A.4 PROPORTIONALITIES

Furthermore, the implications of changing the proportionality of parameter updates may be significant. As stated, both the gradient-only and structural approaches can be further subdivided into proportional and disproportional approaches. These may change the onus of which form of parameter, such as weights or bias, contributes most significantly to the representational corrections. It acts somewhat like the gradient-only preconditioning, yet it uniquely features explicitly in the forward pass. Changing these proportions is felt to be underdiscussed and has several potential ramifications.

It appears some empirical performance is attributable to such changing proportions of the representation correction, due to the empirical support of the $\sqrt{n}$ factor discussed in Zhang & Sennrich (2019). This is of particular interest, as such a scaling can be absorbed into parameters without forward-pass implications, yet was necessitated practically. Thus, its impact remains in the backwards pass, affecting parameter trajectories. The empirical necessity observed by Zhang & Sennrich (2019) may be perhaps the relative up-scaling of weight responsibility counteracting $L_2$-norm's suppression.

This *structural* proportionality consideration remains conceptually intriguing, whether one accepts the affine divergence hypothesis or not — the relative size of corrections applies generally. These proportions can be altered whilst continuing to resolve the affine-divergence, allowing a mode through which to test this in future work.

# B  CLARIFICATIONS ON THE NATURE OF NORMALISERS

## B.1  UNIFYING ACTIVATION FUNCTIONS AND NORMALISERS

In this work, often a 'normalisation-like' approach is discussed. However, these implementations are non-parameterised and not equivalent to typical normalisations implemented by libraries. This discussion will examine such choices, treating normalisers not as a single operation but as a two-step decomposition into a parameterised scaling and what amounts to a non-standard activation function.

Normalisations typically include two characteristic components: a parameterised scaling and some form of statistic-based or nonlinear mapping. For example, BatchNorm standardises activations using the elementwise means and standard deviations over the batch, then scales and offsets using the parameterised $\vec{\gamma}$ and $\vec{\beta}$.

The first step typically reduces the *representational* degrees-of-freedom (through the standardisation), $\mathbb{R}^{b \times n} \to \mathbb{R}^{(b-1) \times n} \to \left(S^{b-2}\right)^n \hookrightarrow \mathbb{R}^{b \times n}$. Then the scaling introduces additional *parameter* degrees of freedom. As discussed, this exchange of degrees-of-freedom from representational to parameterised *do not equate*, and can be reformulated as two individual and independent operations composed into one 'normaliser'. Moreover, the parameterisation can, during the forward pass, be absorbed into existing affine parameters, although they diverge in gradient trajectories when separated.

Hence, these are analytically two separate mappings, motivating separate treatment despite their standard composed usage. This work focuses on the non-parameterised step to draw equivalences with the affine correction. This is why BatchNorm/LayerNorm/RMSNorm are not parameterised in the methodology to enable fair comparison with such corrections.

For affine layers with a norm, the change in the number of representational degrees of freedom is as follows. From this accounting, one can see that the activation degrees of freedom are lost for all norms, but not for the affine correction.

- No norm: $\mathbb{R}^{b \times n} \to \mathbb{R}^{b \times n}$.
- BatchNorm: $\mathbb{R}^{b \times n} \to \mathbb{R}^{(b-1) \times n} \to \left(S^{b-2}\right)^n \hookrightarrow \mathbb{R}^{b \times n}$.
- LayerNorm: $\mathbb{R}^{b \times n} \to \mathbb{R}^{b \times (n-1)} \to \left(S^{n-2}\right)^b \hookrightarrow \mathbb{R}^{b \times n}$. One can see that, geometrically, this is BatchNorm operating on a different axis — much like a transposed version.
- RMSNorm: $\mathbb{R}^{b \times n} \to \left(S^{n-1}\right)^b \hookrightarrow \mathbb{R}^{b \times n}$.
- $L_2$-Norm: $\mathbb{R}^{b \times n} \to \left(S^{n-1}\right)^b \hookrightarrow \mathbb{R}^{b \times n}$.
- Affine Correction: $\mathbb{R}^{b \times n} \to \left(S^{n-1} \times [0, 1)\right)^b \hookrightarrow \mathbb{R}^{b \times n}$, this does not suppress representational degrees-of-freedom, and acts more like performing a coordinate reparameterisation.

Secondly, the separation of these typically composed steps results in a loss of distinction between normaliser and activation functions, as both produce comparable mappings. One could reestablish such a distinction by normalisers reducing the representational degrees-of-freedom or that they use some form of statistic — although this is felt needless. Instead, their unification can be embraced, and this half of the normalisation step can be treated on equal footing to activation functions and their subsequent impact on representational geometry. Moreover, the distinction blurs further if considering parameterised activation functions. In this picture, BatchNorm becomes a non-trivial activation function over batches, rather than single samples, and RMSNorm becomes conceptually analogous to hyperspherical networks (Mettes et al., 2019) at intermediate layers.

This does not deny the empirical fact that combining these two aspects is successful, and they are usually considered as a whole. Instead, this is argued algebraically, as considering normalisers to be a separable two-step process, which can be decomposed and can be analysed as parts. Hence, one aspect is largely algebraically indistinguishable from activation functions and pertinent to the affine divergence perspective. The other component acts as a genuine parameterised affine map, although with sparsity restrictions, such as a diagonalised weight matrix. The latter is largely dropped in this discussion, as it acts as a confounder, allowing a more comparable ablation test across differing approaches to determine the validity of this divergence hypothesis as a mechanistic route for the success of normalisers.

Hence, the current practice could be described as *normalisation + activation function*; however, it can perhaps be better reframed as *parameterised scaling + activation function + activation function*, then fusing the activation functions and merging the scaling with the affine transforms (although separate for differing gradient descent trajectories). By drawing such algebraic equivalences, the amended design approach can be reframed as the combination *parameterised scaling + activation function* and analysed in this manner moving forward. This clarifies the geometrical implications which may constitute foundational biases.

This unification between activation functions and normalisers is evident from *Fig.* 5, which displays similar results as before, but for classification networks *without* an explicit additional activation function, but instead parameterless normalisers composed with affine layers. Due to their isotropic-like form, classical universal approximation theorems (Cybenko, 1989; Hornik et al., 1989; Hornik, 1991) do not hold, and would need rederivation assuming existence[9].

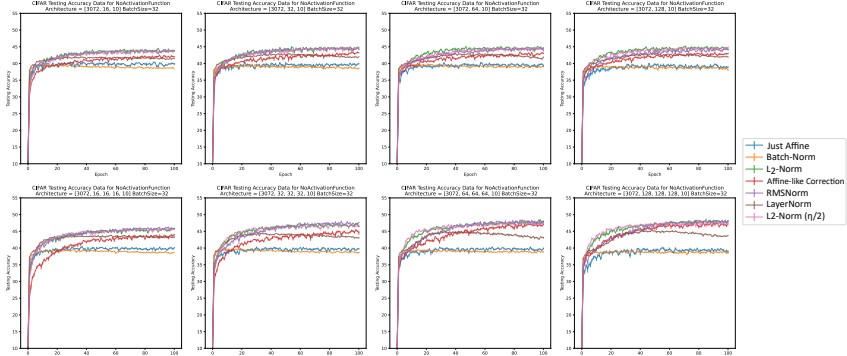

Figure 5: Depicts performance for various networks with no explicit nonlinearity. Titles indicate architecture, with training otherwise identical to *Fig.* 1. These results show that both BatchNorm and no-normaliser perform poorly, and similarly. This is because they are manifestly linear samplewise, so they can achieve at most a linear fit. This is not the case for all other normalisers, which are nonlinear activation functions mathematically and so exhibit nonlinear expressibility. There is a tendency for LayerNorm to be in the middle in performance, perhaps due to its additional losses in representational degrees of freedom. Affine-like improves on this with some variability, then Norm-like, and RMS-like, which often perform similarly well. This may be because the latter two are more classical activation-function-like operations, which are composable with the affine layer, rather than being inseparable as with the affine-like, which is arguably even further from UAT support.

This result indicates that the normaliser decomposition into parameterised scalings and an activation function is a reasonable algebraic equivalence, demonstrating that normalisers display nonlinear expressibility comparable to *Fig.* 1. Analysing normalisers in this decomposed approach may yield further insight, as it facilitates generalisation from modern activation function theory and literature.

Overall, considering this normaliser–activation unification can undermine this tenuous but persistent categorical distinction in these often disparately considered operations. Therefore, their mechanistic and geometric actions may be better considered separately as these two maps, through which activation function theory could be unified. This decomposition has been utilised throughout this paper to enable fair comparison of the nature of the nonlinear/statistical map with the affine divergence corrections.

## B.2   RETAINING THE SCALING CAUSES DIVERGENCE

If one chooses to retain the *parameterised scaling*, then the affine divergence *must* be reconsidered, due to the addition of several new terms in the divergence. This is a higher order in the learning rate and is non-trivial to correct. This does not invalidate conclusions regarding normalisation under

---

[9]However, this is not felt to invalidate their categorisation as activation functions, which necessarily emerged prior to UATs. This lack of UAT applicability, naturally, further indicates a departure from traditional deep learning ontology (Bird, 2025a), particularly sequential affine+elementwise nonlinearity construction.

the original context. But retention of parameterised scalings results in the divergence being only mitigated rather than identically cancelled.

Considering the parameterised rescaling step as an additional affine layer, like those present in BatchNorm and LayerNorm, then when composed with the existing affine layer, it becomes the expression of *Eqn.* 29, with $\tilde{x}$ indicating some preprocessing to $\vec{x}$ — such as BatchNorm's standardisation.

$$y_i = W_{ij}\underbrace{(\gamma_j \tilde{x}_j + \beta_j)}_{z_j} + b_i \qquad (= W_{ij}\gamma_j\tilde{x}_j + W_{ij}\beta_j + b_i) \tag{29}$$

Implementing the gradient-descent step update yields *Eqn.* 30.

$$y_i' = \left(W_{ij} - \eta g_i\gamma_i z_j\right)\left(\gamma_j - \eta g_k W_{kj}\tilde{x}_j\right)\tilde{x}_j + \left(W_{ij} - \eta g_i\gamma_i z_j\right)\left(\beta_j - \eta g_k W_{kj}\right) + \left(b_i - \eta g_i\right) \tag{30}$$

This is an affine layer composed of another affine layer, with a diagonal scaling with respect to the standard basis. This two-layer approximation is much more complicated, as shown with the grouped $\eta^2$ terms of *Eqn.* 31. This expansion displays no clear correction remedy available.

$$\begin{aligned}
y_i' = {}& y_i \\
& - \eta\left(g_i\gamma_j\gamma_i z_j\tilde{x}_j + g_k W_{kj}\tilde{x}_j W_{ij}\tilde{x}_j + g_i\beta_j\gamma_i z_j + g_k W_{kj}W_{ij} + g_i\right) \\
& + \eta^2\left(g_k W_{kj}\tilde{x}_j g_i\gamma_i z_j\tilde{x}_j + g_k W_{kj}g_i\gamma_i z_j\right)
\end{aligned} \tag{31}$$

As stated, this additional diagonalised affine-scaling is separated from normalisation in this paper, predominantly because normalisation itself is two distinct algebraic maps that are composed and can be treated separately for greater clarity. Yet, also because the parameterised step confounds the divergence terms nontrivially. Thus, separating the normaliser's composed scaling steps is a necessary methodological step, but is argued to be generalised since the distinction is largely conventional rather than mathematically fundamental — demonstrating greater algebraic and geometric alignment with activation functions than statistical operators; hence, it should be analysed as such.

## C    DIVERGENCE GENERALISATIONS

So far, this work has shown how the single-sample, first-order, affine divergence may be conceptually and empirically significant for deep learning. However, affine maps are far from the only form of mapping used in modern deep learning; moreover, batching is standard and so far underconsidered in this paper.

Hence, such considerations should be discussed in a more general picture. In this appendix, the implications for batched inputs, convolution, briefly, residuals, and query-key attention are laid out. The derivations for convolution and batched-affine are surprisingly analogous, although different approximation choices may be selected with demonstrated significant empirical consequences.

The (self-)attention approach suggests that such corrections may be intractable in the near term, or at least not a 'cheap' solution to implement. This may indicate why normalisation is not usually paired with a query-key attention step, perhaps because the divergence is not of the same form as that of affine layers; hence, adding a standard normaliser does not provide an advantage through correction.

### C.1    BATCHED INPUTS

In this section, the batched-affine layer will be analysed, including the two proposed structural corrections, affine-like and norm-like, and how these interplay with the batched nature.

Batching is a more complicated consideration, with the resultant representational correction not being perfectly ideal due to the parameter's update being accumulated over several samples. This provides an 'averaged'-out representational correction which does not perfectly satisfy all samples simultaneously. This somewhat relates to the limits of linear maps, preventing all samples from being adapted ideally concurrently.

To proceed, a short overview of the structural corrections are presented thus far.

$$y_i = W_{ij}x_j + b_i \quad \Rightarrow \quad y_i' = y_i - \eta g_i\left(x_jx_j + 1\right) \tag{32}$$

The two solutions are given in *Eqn.* 33 and 34 for affine-like and norm-like structural corrections, respectively (note $t = s^{-1}$).

$$y_i = t\left(W_{ij}x_j + b_i\right) \quad \Rightarrow \quad y_i' = y_i - \eta g_i t^2\left(x_jx_j + 1\right) \quad \Rightarrow \quad t = \frac{1}{\sqrt{x_jx_j + 1}} \tag{33}$$

$$y_i = tW_{ij}x_j + b_i \quad \Rightarrow \quad y_i' = y_i - \eta g_i\left(t^2x_jx_j + 1\right) \quad \Rightarrow \quad t = \frac{1}{\sqrt{x_jx_j}} = \frac{1}{\|\vec{x}\|_2} \tag{34}$$

Now the batched generalisation can be explored, followed by what each of these approaches induces on the batch-wise picture.

A batched-affine map can be summarised as *Eqn.* 35, where $b$ indexes the batch.

$$y_{bi} = W_{ij}x_{bj} + 1_b b_i \tag{35}$$

The respective gradients are given by *Eqns.* 36

$$\frac{\partial \mathcal{L}}{\partial y_{nm}} = g_{nm} \qquad \frac{\partial \mathcal{L}}{\partial W_{nm}} = g_{kn}x_{km} \qquad \frac{\partial \mathcal{L}}{\partial b_n} = g_{kn}1_k \tag{36}$$

One can see that there is now a gradient matrix, rather than a vector, with respect to $y_{nm}$. For representations, this batchwise index is preserved, whereas for both $W_{nm}$ and $b_n$, a contraction automatically accumulates the batchwise gradients, according to the accumulation specified by $\mathcal{L}$, such that the batch is no longer expressed in their gradient. This is because the activations carry sample-specific information; in contrast, parameters are reused across all batches. A perhaps unusual restatement of this is that there is weight sharing across samples, paralleling convolution.

Additionally, the accumulation of gradients over samples for the parameters is typically linear, e.g. a mean, derived from loss. This is a subtly different consideration from the clear linear index contractions in the backpropagation step. This is because the terms *can* be evaluated non-linearly even if superficially linearly index-contracted

Substituting these in as a gradient descent step and propagating to a representational update yields *Eqn.* 37.

$$
\begin{aligned}
y'_{bi} &= y_{bi} - \eta \left( g_{ki} x_{kj} x_{bj} + g_{ki} 1_b 1_k \right) \\
y'_{bi} &= y_{bi} - \eta g_{ki} \underbrace{\left( x_{kj} x_{bj} + 1_{bk} \right)}_{= \mathbf{M}_{bk}} \\
y'_{bi} &= y_{bi} - \eta \mathbf{M}_{bk} g_{ki}
\end{aligned}
\tag{37}
$$

We can observe here a more complicated affine divergence $g_{bi} \neq \mathbf{M}_{bk} g_{ki}$, unless a scaled identity matrix forms: $\mathbf{M}_{bk} = \mathrm{I}_{bk} (= \delta_{bk})$ (requiring differing samples to all simultaneously have an inner product of $-1$). However, in general, a gram-like matrix divergence arises batchwise. This causes a sample-wise linear mixing of the gradient $\mathbf{M}_{bk} g_{ki}$, which is expected given the parameter updates with respect to the batch-accumulated gradients.

An interpretation is that the representation correction is a somewhat (weighted-)average over the batch. This also means that not all steps are often simultaneously ideal. Without structural corrections, this weighted sum may be far from the ideal step for representations, e.g. $g_{bi}$.

One could experiment with a correction factor, e.g. $y_{bi} = t_{kb} \left( W_{ij} x_{kj} + 1_k b_i \right)$, but this requires a (pseudo-)inverse square-root matrix, which is computationally very expensive. Pseudo-inversion is especially needed when the index length of $j$ is not equal to that of $b$, or when there is rank deficiency. Furthermore, this would also mix batch information non-trivially and likely undesirably for most forward-pass applications.

Instead, we can begin by investigating the two structural corrections, which will elucidate their implications on the batched picture. The two approaches yield corrections *Eqns.* 38 and 39 for affine-like and norm-like, respectively.

$$
y'_{bi} = y_{bi} - \eta g_{ki} \frac{x_{kj} x_{bj} + 1_{bk}}{s_b s_k} \quad : \quad s_n = \frac{1}{\sqrt{x_{nm} x_{nm} + 1_n}}
\tag{38}
$$

$$
y'_{bi} = y_{bi} - \eta g_{ki} \left( \frac{x_{kj} x_{bj}}{s_b s_k} + 1_{bk} \right) = y_{bi} - \eta g_{ki} \left( \hat{x}_{kj} \hat{x}_{bj} + 1_{bk} \right) \quad : \quad s_n = \frac{1}{\sqrt{x_{nm} x_{nm}}}
\tag{39}
$$

Two immediate conclusions are that the diagonal terms dominate due to these two corrections. This is conceptually notated in *Eqn.* 40, with $\mathbf{N}_{bk} = \left( 1_{bk} - \delta_{bk} \right) \mathbf{M}_{bk}$ or equivillantly $\mathbf{M}_{bk} = \delta_{bk} + \mathbf{N}_{bk}$, which preserves only the off-diagonal entries of $\mathbf{M}_{bk}$. Moving forward $\eth_{ij} = \left( 1_{bk} - \delta_{bk} \right)$ will indicate the off-diagonal ones, e.g. $\eth_{ij} + \delta_{ij} = 1_{ij}$

$$
y'_{bi} = y_{bi} - \underbrace{\eta' g_{bi}}_{\text{Ideal}} - \underbrace{\eta' \mathbf{N}_{bk} g_{bk}}_{\text{Effective Interference}} \approx y_{bi} - \eta' g_{bi} \quad \text{since} \quad |\mathbf{N}_{bk}| \leq 1_{bk}
\tag{40}
$$

Therefore, the change in $y_{bi}$ is most significantly *weighted* towards $g_{bi}$, and therefore, when assuming i.i.d. samples and with similar magnitude $g_{bi}$, will tend to dominate the correction for the sample. This is the desired ideal update step, encouraged by the two structural corrections. This is followed by a sum of smaller magnitude of 'interference' corrections $\mathbf{N}'_{bk} g_{ki}$, due to other sample gradients in the off-diagonal terms. Secondly, the persistent $1_{bk}$ generally adds a weighting to off-diagonal terms unless $\hat{x}_{kj} \hat{x}_{bj} = -1$. Hence, these are all positively weighted sums of $g_{ki}$.

Denoted more directly, *Eqns.* 41 and 42 demonstrate the diagonal and off-diagonal decomposition explicitly.

$$
y'_{bi} = y_{bi} - \eta g_{bi} - \eta g_{ki} \left( \eth_{bk} \frac{x_{kj} x_{bj} + 1_{bk}}{\sqrt{x_{bm} x_{bm} + 1_b} \sqrt{x_{km} x_{km} + 1_k}} \right)
\tag{41}
$$

$$y'_{bi} = y_{bi} - 2\eta g_{bi} - \eta g_{ki} \left( \eth_{bk} \left( \hat{x}_{kj} \hat{x}_{bj} + 1_{bk} \right) \right) \tag{42}$$

In both cases, the leading weighted correction then becomes $\propto g_{bi}$ as desired, approximating the affine correction. Other terms then produce slight further geometric propagated corrections, weighted by this off-diagonal sample-mixing matrix.

If a particularly large $g_{ki}$ is present, then this may adversely affect the result, particularly if it overpowers the $\mathbf{N}_{kb}$ suppression. This may lead to a poorer performing sample before and after update, but generally $g_{bi}$ should probabilistically lead due to the favoured weighting.

One could explore bounding $g$'s magnitudes, e.g. norm clipping, such as to suppress this eventuality. Exploring whether this generally aids performance by making all samples contribute an equal magnitude of correction. This is analogous to suppressing $\|\vec{x}\|^2 + 1$ throughout this work, which may be an interesting separate direction that links this paper's exposition to norm-clipping techniques.

Overall, heuristically, we may expect that the norm-like and affine-like structural corrections will produce more ideal gradient steps for representations and contribute an overall benefit samplewise, by suppressing off-diagonal terms — at least if these off-diagonal steps are not drastically damaging to the overall correction.

### C.1.1 AUXILIARY BATCHED HYPOTHESIS

Most importantly, the dominant diagonal ideal correction with a *sum* of off-diagonal sample-mixing interference can be interpreted to provide another falsifiable auxiliary hypothesis for this theory.

As the batch size increases, there is no fixed norm for the contribution of interfering terms; therefore, with a greater number of samples, we would expect greater interference to occur. This results in a geometric deflection away from the 'ideal' representational update per sample.

As a direct consequence, one could predict that with respect to this theory: *a greater number of samples should directly **degrade** the efficacy of the structural corrections.*

This would not be the case for other normalisers, which may function differently in the representation alignment picture. For functions, such as BatchNorm, we may expect that an increasing number of samples would better produce better statistics (conventional picture) or better approximate damping of the $\mathrm{Var} \|\vec{x}\|^2$ (ideal misalignment picture), reducing the impact of individual detrimental fluctuations over the accumulated batch, potentially improving performance. A similar argument can be made for no normaliser, where batches stabilise gradient updates when accumulated (to some degree, this may also counteract the negative correlation hypothesised from the theory).

This simple hypothesis can be trivially tested by comparing performance across batch sizes for various normalisers, as shown in *Fig.* 6 for standard-tanh and *Fig.* 7 for Leaky-ReLU. These are for 2 hidden layers with a width 32 for classification on CIFAR-10.

One can see immediately that the two structural corrections have a relation where increasing batch size negatively correlates with performance for batched-affine layers — this is not the case for other prior normalisers. These correlations are tabulated in *Tab.* 1.

| Normaliser | Standard Tanh | | Leaky ReLU | |
| --- | --- | --- | --- | --- |
| | Avg. Acc. | Slope | Avg. Acc. | Slope |
| Just Affine | $38.35 \pm 0.50$ | $(5.80 \pm 1.17) \times 10^{-2}$ | $47.54 \pm 0.01$ | $(7.33 \pm 2.81) \times 10^{-3}$ |
| BatchNorm + Affine | $39.08 \pm 0.87$ | $(5.04 \pm 1.21) \times 10^{-2}$ | $43.84 \pm 0.44$ | $(3.13 \pm 0.96) \times 10^{-2}$ |
| LayerNorm + Affine | $43.72 \pm 0.21$ | $(-8.77 \pm 2.45) \times 10^{-3}$ | $45.57 \pm 0.02$ | $(-5.32 \pm 3.27) \times 10^{-3}$ |
| RMSNorm + Affine | $27.31 \pm 1.87$ | $(1.05 \pm 0.21) \times 10^{-1}$ | $48.80 \pm 0.02$ | $(-8.20 \pm 1.68) \times 10^{-3}$ |
| $L_2$-Norm + Affine $(\eta)$ | $49.93 \pm 0.10$ | $(-2.98 \pm 0.20) \times 10^{-2}$ | $48.64 \pm 0.02$ | $(-5.75 \pm 2.74) \times 10^{-3}$ |
| $L_2$-Norm + Affine $(\eta/2)$ | $48.42 \pm 0.35$ | $(-2.58 \pm 0.53) \times 10^{-2}$ | $48.73 \pm 0.002$ | $(-6.75 \pm 0.93) \times 10^{-3}$ |
| Affine-Like Correction | $\mathbf{50.56 \pm 0.13}$ | $(-2.45 \pm 0.17) \times 10^{-2}$ | $\mathbf{49.26 \pm 0.01}$ | $(-3.39 \pm 1.07) \times 10^{-3}$ |

Table 1: Displays the average accuracy of networks on CIFAR-10 taken across various batch sizes, alongside the slope of accuracy against batch size for these networks. Blue indicates negative slopes, and emboldening indicates the highest accuracy. These are taken by fitting a linear regression to the data points.

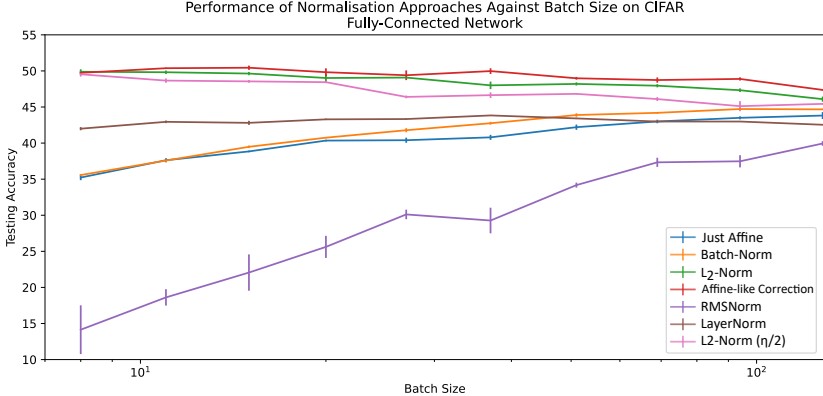

Figure 6: Displays the CIFAR10 test set accuracy for a batched-affine classification network using standard tanh, against differing batch sizes. Batch sizes range from 8 to 128 samples per batch, and trained for over 100 epochs. One can see that two structural corrections, "$L_2$-Norms" ($\eta/2$ and $\eta$ — all other experiments using $\eta = 0.001$) and "Affine-like Correction", negatively correlate final accuracy with batchsize. This contrasts with LayerNorm, which is approximately constant, and BatchNorm, Standard Affine (e.g., no normalisation), and RMSNorm, which all positively correlate. Across all cases, the three structural corrections outperform all other approaches. These results are also for de-parameterised normalisers consistent with the discussion in *App.* B; which accounts for any seemingly non-standard normalisation results (i.e. low performance of RMSNorm). Error bars indicate standard error, equivalent to the error on the mean statistic.

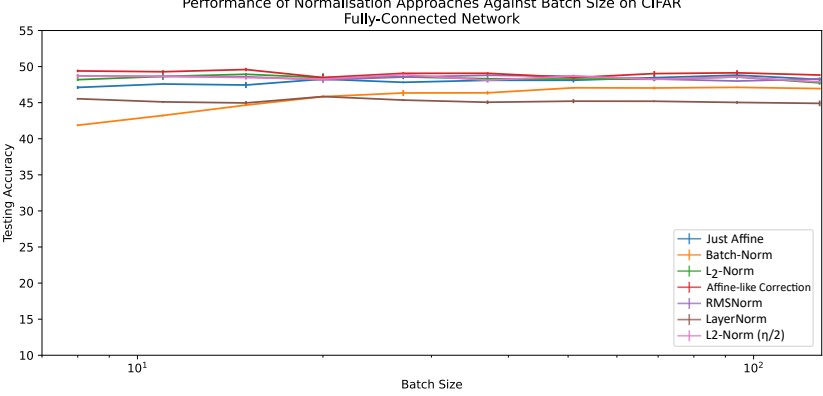

Figure 7: Displays the CIFAR-10 test set accuracy for a batched-affine classification network using Leaky-ReLU, against differing batch sizes ranging from 8 to 128 samples per batch, over 100 epochs. Mean and standard error are shown. Similar to previous results, these performances are more comparable and difficult to separate. Across all samples, the affine correction is the best-performing and exhibits a slight negative correlation.

Across all cases, the affine-like correction outperforms other normalisers and is consistent with prior results. Moreover, all affine-like slopes, $L_2$-Norm (norm-like correction) at half and full $\eta$ have a negative correlation with batch size. This clearly demonstrates that all structural corrections follow the expected and counterintuitive declining performance with batch size. These are also statistically significant results, with only Leaky-ReLU $L_2$-Norm ($\eta$) and affine-like having slightly larger errors.

Generally, for Leaky-ReLU, all slopes are significantly smaller due to the more comparable results. LayerNorm also exhibits a negative slope, but the theory does not make *any* predictions regarding LayerNorm, so minimal interpretation is possible — similar for RMSNorm's negative slope for Leaky-ReLU. Their respective interference are not a sum of strictly positive terms due to rescaling. The errors on Layer Norm's terms are also proportionally larger.

Overall, this appears to strongly support the auxiliary hypothesis and overall misalignment theory, showing a surprising result: greater batch sizes tend to negatively correlate with structural corrections, which may be counterintuitive and classically surprising.

The RMSNorm and similarly LayerNorm can be analysed through their representation updates featuring a similar rescaling shown in *Eqn.* 43, where $\vec{x} \in \mathbb{R}^{B \times n}$.

$$y'_{bi} - y_{bi} = -\eta g_{ki} \left( \mathbf{n} \hat{x}'_{kj} \hat{x}'_{bj} + 1_{kb} \right) \quad : \quad \vec{x}'_{bj} = \begin{cases} \vec{x}_{bj} - \hat{1}_{bk} \vec{x}_{bk} \hat{1}_{bj} & : \quad \text{LayerNorm} \\ \vec{x}_{bj} & : \quad \text{RMSNorm} \end{cases} \tag{43}$$

The $n(=32)$ weighting in the correction would imply an additional onus on the weights for providing the representational correction, seen especially if $\eta \to \eta/(n+1)$. Effectivly, this a redistribution of responsibility for the representation update between parameters. The summation of weighted $g_{ki}$ is more complicated in such a picture, with the presence of potentially large negative weightings, although with a remaining dominant diagonal and a relatively much larger amplification of cross-sampling coupling. Due to this, it is presently unclear how the sum of these positive and negative off-diagonal terms may interfere, preventing an analytical prediction for the performance relation with batch size.

Perhaps, with a greater batch size, more positive and negative contributions could also tend to cancel out or randomly compound, forming an expected distribution — the latter would also suggest negative correlation. Shifting parameter responsibility may also be impactful. Additionally, in larger samples, the various $\vec{x}$ may spread more evenly across the available volume, leading to more effective cancellation of both positive and negative cross-terms. In contrast, in positive-only $L_2$, these contributions grow in effect.

In any case, determining such an effect for both LayerNorm and RMSNorm is unclear, and not directly hypothesised as a result of the ideal-effective misalignment at this stage. Hence, their results should not be used to in/validate the theory, since the prediction of decreasing performance pertains only to the $L_2$-norm, which is empirically validated. Further research may better elucidate RMSNorm and LayerNorm's dependence. Nevertheless, the batchwise mixing term $M_{bk}$ is present only in the *representation*-propagated corrections and downstream, suggesting it is implicated in these observations to some degree.

BatchNorm's trend, as mentioned, appears consistent with the damping/removal of $\text{Var} \|\vec{x}\|^2$ by confining the distribution via batch statistics, and also convergence by batch averaging. Thus it acts as a mitigation to the affine divergence, not an identical cancelling of terms. This mitigation improves as $\text{Var} \|\vec{x}\|^2 \to 0$, suggestive of a positive batchsize-performance correlation in the representation update picture. Other classical modes for BatchNorm's success may also be pertinent.

Overall, the results support that the cumulative interference from off-diagonal terms does grow with batch size, impacting the overall accuracy. This appears to produce the surprising and non-standard negative correlation with batch size for the structural corrections. These considerations aid in isolating this mode as causal. This is an entirely distinct secondary falsifiable hypothesis that further strongly supports the ideal-effective misalignment theory, not merely as an incidental but *a* predictive mechanistic explanation of normalisation's empirical success. This, in addition to evidence that the affine-like correction is successful despite not being a typical normaliser, suggests that this affine divergence theory may be significant. Such a result appears at least interesting with respect to the role of the affine divergence in the empirical success of normalisation, and at minimum it is suggested to warrant further investigation.

### C.1.2 The Sample Individuality Decision

The two structural corrections are mathematically shown to weight the sample-wise representational correction towards each sample's ideal gradient step. From an individual sample perspective, this appears beneficial. Yet, from the batched perspective, this remains non-ideal with the possibility of

further improvement — although to implement this is computationally prohibitive and conceptually undesirable to create general interdependency between samples on the forward pass[10].

Hence, these decisions regarding structural corrections entail an implicit prioritisation of sample individuality. The choice is made not to implement batchwise coupling, which might *in principle* yield a lower aggregate loss in the affine-divergence picture; instead, each sample is engineered such that the leading term is its ideal gradient, which may be most important for individually reducing that sample's overall loss but not the ensemble due to the residual batch-divergence.

It suggests that these updates are still effective at reducing the loss individually; the overall batch loss is then straightforward to accumulate from these individual samples: a (linear) mean combination.

Consequently, the gradient $g_{bi}$ has a sort of implicit linearity in $b$, despite its strong non-linearity in $i$'s relations (e.g. the only cross terms in $b$ arise linearly in the loss, this is not so for $i$). This makes each sample's drop in loss correspond to a decline in the batch loss due to the straightforward linear combination of individual samples in the loss (neglecting spurious increases samplewise from $\bar{\eth}_{kb}\mathbf{M}_{kb}g_{ki}$). This is an implicit choice of sample-independence made as an approximation, and it is seemingly heuristically favourable in the batch picture.

This consideration does not apply if all samples are non-linearly mixed in the forward pass to yield a final outcome. Despite the superficial linear contraction of $g_{bi}$, the evaluation of $g_{bi}$ becomes very non-linear with respect to the batched-index of $x_{bi}$, and such a motivation of sample-wise independent prioritisation no longer holds. In this case, the presence of non-ideal off-diagonal terms may be much more damaging overall, even if it offers *some* improvement upon no normalisation, although even this may be doubtful.

This is conceptually significant. As in the subsequent section, it will be demonstrated, perhaps surprisingly, that the batched-affine and convolutional are structurally identical in the ideal-effective gradient picture, where patches act like the batched index. In short, batched affine layers are much like convolving the network sample-wise. Despite the unexpected parallel, these patches are not independent and are not accumulated linearly in the loss, as individual samples are. Hence, despite the structurally identical divergence, the partial structural corrections do not offer a straightforward solution.

The following subsection discusses this convolutional case and why such structural corrections are not clear, despite superficially being identical to the batched divergence.

## C.2   CONVOLUTIONAL DIVERGENCE

Qualitatively, the divergence occurs similarly for convolutional layers, but implementation-wise, the correction diverges strongly from current practice, suggesting a Patch-wise consideration over layer/group/batch-wise.

Potential solutions to a convolutional divergence are again numerous; yet, several will be showcased. Particularly, a novel approach to the application of normalisation in convolution is presented, to be termed "*PatchNorm*", which, unlike the pre/post-composition of normalisation usually accompanying convolution, PatchNorm is an intrinsic change to the convolution procedure itself. "PatchNorm" is not to be taken as *a* function, but a family of normalisation applied patch-wise.

First, a parallel will be drawn with the discussion of *App.* C.1, requiring reformulating the typical convolution expression shown in *Eqn.* 44.

$$y_{ijd} = W_{abcd}x_{(i-a)(j-b)c} + b_d \tag{44}$$

Equivalently in function, one can 'unroll' convolution to express 'patches' as an index matrix $x_{ijc} \rightarrow \tilde{x}_{pe}$, where $p$ indexes each patch per sample, and $e$ indexes the dimensions per patch. Notably, $\tilde{x}$ contains many repeated instances of the original $x_{ijc}$ elements due to the patchwise unfolding. Overall, with corresponding reshaping of $W$, convolution becomes equivalent to *Eqn.* 45.

---

[10]This interdependence of samples is substantively different from the estimated batchwise statistics stored and used in BatchNorm at inference, instead in this picture, such running estimates are not possible as a sample-dependent coupling is needed to be computed every time for the current specific samples.

$$\tilde{y}_{pd} = 1_p W_{ed} \tilde{x}_{pe} + 1_p b_d \tag{45}$$

Comparing this to *Eqn.* 35 one can see that these two operations are structurally identical in terms of matrix algebra (differing only in the $1_p$, which is made explicit in convolution to indicate the patchwise broadcasted weights, it is implicit in *Eqn.* 35). Thus, the convolutional divergence is shown in *Eqn.* 46.

$$\tilde{y}'_{bi} = \tilde{y}_{bi} - \eta \left( 1_p 1_k g_{kd} \tilde{x}_{ke} \tilde{x}_{pe} + g_{kn} 1_k 1_p \right)$$
$$\tilde{y}'_{bi} = \tilde{y}_{bi} - \eta g_{kd} \left( \tilde{x}_{ke} \tilde{x}_{pe} + 1_{kp} \right) \tag{46}$$

Hence, the divergence is also structurally *identical* to the case for batched affine. However, notable crossover in patched inputs does occur with instances of repeated value in $\tilde{x}$ induced by the unrolling — this is unlike the batched case *in general*. Similarly, this equation could be expressed with an additional batched index $b$, but structurally this does not cause a different divergence as it can be equally absorbed into a flattening and reindexing of $p$ — so is notationally neglected for simplicity (although differing by instead inheriting the single-sample assumptions of *App.* C.1).

Of course, one could still implement the same structural corrections for empirical confirmation, yielding two new forms of deep learning map "PatchNorm" for Patch-Normalised-Convolution. These new 'normalisers' are inseparable from convolution, in contrast to typical normalisation functional forms that can be used in composition. Hence, two primary forms of PatchNorm can be introduced, in complete analogy to the affine corrections, *Eqns.* 47 and 48, analogous to affine-like and norm-like approaches respectively.

$$\tilde{y}_{pd} = t_p \left( 1_p W_{ed} \tilde{x}_{pe} + 1_p b_d \right) \quad \Rightarrow \quad t_b = \frac{1}{\sqrt{\tilde{x}_{ba} \tilde{x}_{ba} + 1}} \tag{47}$$

$$\tilde{y}_{pd} = \left( t_p W_{ed} \tilde{x}_{pe} + 1_p b_d \right) \quad \Rightarrow \quad t_b = \frac{1}{\sqrt{\tilde{x}_{ba} \tilde{x}_{ba}}} \tag{48}$$

Yet, for convolution, the underlying approximations used to justify these single-sample corrections in batched cases are fundamentally broken by the non-linear mixing of individual patches to produce the output. This means that the approximations are not justified for convolution.

Therefore, despite the identical arithmetic to that of batched affine, the breakdown of a core approximation makes it unclear whether these implementations are supported. There are also strong inductive-bias arguments against perfectly correcting for the divergence, as it interferes with locality.

Overall, despite their structural appearance, patches are not comparable to samples in nature. Treating batched samples independently, neglecting batch-coupling corrections, is considered a defensible approximation and is consistent with how the loss aggregates individual samples linearly. They are treated individually and equally important in correction, each acquiring a $1/n$ weight in the loss. On average, individually aligning one sample's representational update is expected to reduce that sample's loss more than optimal parameter steps alone. This individual reduction is then linearly impactful on the overall batch's loss as discussed.

Such a case for convolution is not clear. Patches do not contribute independently or linearly to the loss; they are repeatedly non-linearly combined to yield the final loss for a single sample.

Moreover, the $N - 1$ cross-sample interference terms for single-sample approximation, treated as more negligible, become the quadratically greater $N(N-1)$ cross-patch interference terms for a single sample patchwise — all contributing to the same single sample's loss. Hence, it is likely that these are not negligible per sample but considerable in their cumulative effect, thereby diluting the benefit of the structural correction's diagonally aligned gradients for patches.

In this case, the effective mixing appears as $\mathbf{M}_{pk} = \tilde{x}_{ke} \tilde{x}_{pe} + 1_{kp}$, which could be counteracted using a pseudo-root-inverse of the patchwise Gram-like divergence on the forward pass; however this is computationally prohibitive and disrupts the spatialised nature of convolution. This intertwines patches non-locally, directly violating the foundational inductive-bias considerations that motivate

convolution. It also breaks the translational[11] equivariance by introducing inter-patch mixings. Contrary to common belief, this is a less troublesome disruption by structural corrections, since edge effects of standard convolution break it anyway.

Nevertheless, the desirable locality of convolution does not hold. Hence, the network may be sensitive to patchwise mixing, and overall, this is conceptually unappealing. Incidentally, the often successful BatchNorm respects locality, as does the element-wise form of activation functions typically used in convolution — perhaps suggestive of BatchNorm's empirical success in convolution.

Overall, both the prior straightforward structural corrections for batched-affine can be applied individually to patches, but their use is unsupported due to approximation breakdown due to the strong nonlinear computations interconnecting patches downstream. Similarly, full corrections are unfavourable because they undermine the inductive biases of convolution. Therefore, despite remedies for the ideal-effective misalignment theoretically existing and directly comparable to batched-affine corrections, they may be troublesome due to pathological interactions arising downstream. Hence, this requires substantial future study.

Nevertheless, one can still implement PatchNorm as shown in *Figs.* 8 and 9. These are insightful examples, as despite PatchNorm being structurally identical to the batched-affine case, the implicit assumptions fundamentally differ, and subsequent outcomes are markedly different.

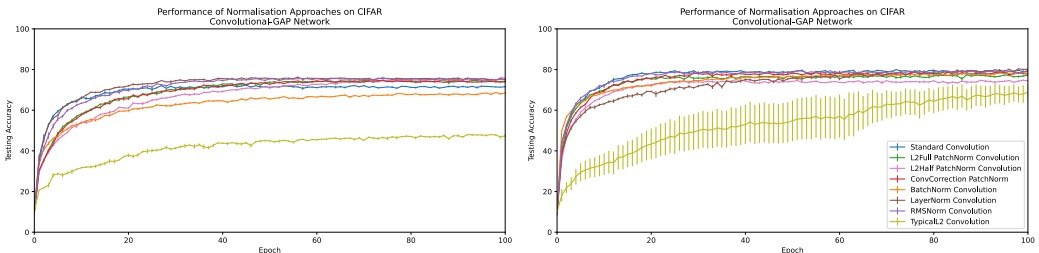

Figure 8: Shows the testing accuracy of a convolutional network, using a global average pool layer, trained for 100 epochs on CIFAR-10. Shown is the mean and standard error on accuracy for various normalisers across 5 repeats. The left plot displays results for Tanh, whilst the right plot displays results for Leaky-ReLU. For Tanh, all PatchNorm variants, RMSNorm and LayerNorm perform comparably well, followed by wider margins no normalisation then BatchNorm and considerably lower the layerwise application of $L_2$ (which in convolutional circumstances, layerwise application is not a divergence correction). For Leaky-ReLU, results are much closer in value, with the highest appearing as LayerNorm, then RMSNorm, then No-Norm, then BatchNorm and the affine-like PatchNorm equivalent (red), but all are closely grouped. Then follow the $L_2$-like PatchNorms, and last by a wider margin is the layerwise $L_2$-Norm. These are all parameterless formulations for ablation testing, which may account for non-standard results.

The results indicate that, despite being arithmetically identical to the affine structural solutions, PatchNorm's performance is not largely superior to that of other normalisers like the previous wide margin — instead performing well but more comparable.

Overall, over the 160 networks tested, with 32 unqiue convolutional networks, the results show that PatchNorm remains successful, but more comparable to existing normalisers, showing that other norms can slightly outperform it.

This may be surprising since, superficially, PatchNorm is identical to the affine divergence correction, which was previously shown to work well by a large margin.

This likely indicates the breakdown of the single-sample-like approximation, more appropriately the single-patch approximation for convolution, which held over prior batched affine experiments. Hence, such a result may also be conceptually significant.

This suggests that these interactions cannot be neglected in approximation, a suggestion directly supported by the empirical results. As discussed, this is likely due to either non-linear mixing of

---

[11]Really more like a special subset of permutations with respect to neurons *not* data. As 'translation' in this sense is more like a reindexing as opposed to translating the activation value $\vec{a}' = \vec{a} - \vec{t}$.

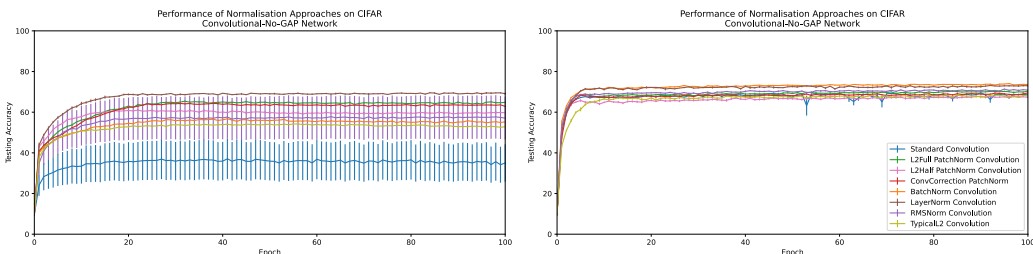

Figure 9: Shows the testing accuracy of a convolutional network, without using a global average pool layer, but instead gradually spatially reducing the activations into channels. This network is trained identically to that in *Fig.* 8, differing only in its architecture. This plot also shows the mean and standard error across. The left plot displays results for Tanh, whilst the right plot displays results for Leaky-ReLU. The tanh results feature better separation compared to other results. LayerNorm is optimal, followed by the three forms of PatchNorm, then with significant variability, RMSNorm, then Batchnorm, then layer-wise $L_2$-Norm, then with a substantial drop to the variably performing No-norm. For Leaky-ReLU, this differs, with BatchNorm displaying optimality, followed by LayerNorm, then RMSNorm, then the two full $\eta$ PatchNorms, with the other PatchNorm, NoNorm and layerwise $L_2$-Norm performing similarly. The slight difference between the two $L_2$-PatchNorms suggests some sensitivity to the learning rate, and this can be optimised for better performance.

patches, repeated numerical values in the unfolding, or differences in backpropagated gradients from activation to activation arising from repeated unfolding values. The explanation may be one of or a combination of these factors, as these delineate how convolution differs from the batched-affine success.

Whether this interpatch dependency also explains why convolution *typically* empirically favours BatchNorm over other normalisers, and why this differs from affine networks, may be explored in this multi-patch ideal-effective consideration, as BatchNorm may offer a better mitigation here perhaps due to $\operatorname{Var}\|\vec{x}\|^2$ damping.

From *Fig.* 8 and 9's foundation, empirically validating the breakdown of the patch-interdependence assumption, one could explore other modes for ideal-effective alignment in representation updates for convolution. This may yield alternative, perhaps empirically more successful, normalisers using this approach. Additionally, such an effort would provide an independent route to verifying or falsifying this ideal-effective update principle for the success of normalisation.

Overall, it appears that generalising the ideal-effective misalignment theory to convolution may require substantial further, nuanced considerations to construct functions that produce approximations that better respect the construction of convolutional layers, and this is encouraged for future work. However, the comparable success of PatchNorm does lend some further support to the ideal-effective misalignment theory, showing its ability to generalise to other architectures, even if PatchNorm does not adequately resolve the divergence.

Secondly, this may not be a fundamental weakness of the novel functional form of PatchNorm, which may be improved with additional parameters, such as those used in traditional normalisers. The two approaches to PatchNorm may continue to have use cases where they are beneficial, and a general investigation of the functional form, e.g., Patchwise normalisation, is strongly encouraged as an alternative to explore compared to existing forms.

In conclusion, despite the convolution case being structurally identical to the batched-affine case, the interpretation is significantly different due to the presence of a single sample and downstream patchwise nonlinear interdependencies, which undermine the structural solutions previously found successful in analogous batched-affine tests. This is partially borne out in the experiments, where structural corrections do largely outperform alternative normalisers, but instead are more comparable — reinforcing that patches are not independent samples. Inter-patch considerations are highly non-linear and cannot be neglected without global consequences for performance. This motivates a future search for convolutional structural corrections that more accurately account for the ideal-effective misalignment to assess the validity of this mechanism.

## C.3 ATTENTION DIVERGENCE

Taking the correlation step of attention to be the expression in *Eqn.* 49, with $\mathbf{X} \in \mathbb{R}^{t \times d}$ being the stacked token vectors, $\mathbf{W}^{(Q)} \in \mathbb{R}^{n \times d}$ and $\mathbf{W}^{(K)} \in \mathbb{R}^{n \times d}$ being the query and key matrices, with $\mathbf{Y} \in \mathbb{R}^{t \times t}$ respectively. Downstream activation functions, 'value' term and multiple heads are suppressed for notational simplicity but trivially extendable from the equations below.

$$Y_{ts} = W_{ip}^{(K)} W_{iq}^{(Q)} X_{tq} X_{sp} \tag{49}$$

From these the respective gradients can be computed with $\frac{\partial \mathcal{L}}{\partial \mathbf{Y}} \equiv \mathbf{g}$.

$$\frac{\partial \mathcal{L}}{\partial W_{nm}^{(Q)}} = g_{ts} W_{np}^{(K)} X_{tm} X_{sp} \tag{50}$$

$$\frac{\partial \mathcal{L}}{\partial W_{nm}^{(K)}} = g_{ts} W_{nq}^{(Q)} X_{tq} X_{sm} \tag{51}$$

$$\frac{\partial \mathcal{L}}{\partial X_{nm}} = g_{ns} W_{ip}^{(K)} W_{im}^{(Q)} X_{sp} + g_{tn} W_{im}^{(K)} W_{iq}^{(Q)} X_{tq} \tag{52}$$

$$\tag{53}$$

Now we can substitute these as a gradient descent update step to determine the effective update on $\mathbf{X}$.

$$Y'_{ts} = \left( W_{ip}^{(K)} - \eta g_{de} W_{if}^{(Q)} X_{df} X_{ep} \right) \left( W_{iq}^{(Q)} - \eta g_{ab} W_{ic}^{(K)} X_{aq} X_{bc} \right) X_{tq} X_{sp} \tag{54}$$

Expanding and simplifying yields, and using $W_{nm}^{(Q2)} = W_{in}^{(Q)} W_{im}^{(Q)}$ and $W_{nm}^{(K2)} = W_{in}^{(K)} W_{im}^{(K)}$ and $X_{nm}^{(g2)} = g_{ab} X_{an} X_{bm}$, yields *Eqn.* 55, the 'attention divergence'.

$$Y'_{ts} = Y_{ts} - \eta \left( W_{qf}^{(Q2)} X_{fp}^{(g2)} + W_{pc}^{(K2)} X_{cq}^{(g2)} \right) X_{tq} X_{sp} + \eta^2 \left( W_{ic}^{(K)} W_{if}^{(Q)} X_{fp}^{(g2)} X_{qc}^{(g2)} \right) X_{tq} X_{sp} \tag{55}$$

We can compute the solution, which must be normalisation-like due to the lack of bias, given by *Eqn.* 56. However, this can be applied, vector-wise, $s_i$ or $s_p = s_q$, token-wise, $s_t = s_s$ or globally, $s$. Proceeding globally with scalar $s$.

$$Y_{ts} = \frac{W_{ip}^{(K)} W_{iq}^{(Q)} X_{tq} X_{sp}}{s} \tag{56}$$

Recomputing gradients yields:

$$\frac{\partial \mathcal{L}}{\partial W_{nm}^{(Q)}} = \frac{g_{ts} W_{np}^{(K)} X_{tm} X_{sp}}{s} \tag{57}$$

$$\frac{\partial \mathcal{L}}{\partial W_{nm}^{(K)}} = \frac{g_{ts} W_{nq}^{(Q)} X_{tq} X_{sm}}{s} \tag{58}$$

$$\frac{\partial \mathcal{L}}{\partial X_{nm}} = \frac{g_{ns} W_{ip}^{(K)} W_{im}^{(Q)} X_{sp} + g_{tn} W_{im}^{(K)} W_{iq}^{(Q)} X_{tq}}{s} \tag{59}$$

$$\tag{60}$$

Then substituting yields *Eqn.* 61

$$Y'_{ts} = Y_{ts} - \eta \left( \frac{W^{(Q2)}_{qf} X^{(g2)}_{fp} + W^{(K2)}_{pc} X^{(g2)}_{cq}}{s^2} \right) X_{tq} X_{sp} + \eta^2 \left( \frac{W^{(K)}_{ic} W^{(Q)}_{if} X^{(g2)}_{fp} X^{(g2)}_{qc}}{s^3} \right) X_{tq} X_{sp}$$

(61)

Then one can attempt a solution for $s$ under various assumptions. In this approach, it will be assumed that $\eta^2 \to 0$ and using the ideal gradient of $g_{ts}$.

$$g_{ts} = -g_{kl} \frac{W^{(Q2)}_{qa} X_{tp} X_{sq} + W^{(K2)}_{pa} X_{tq} X_{sp}}{s^2} X_{ka} X_{lq}$$

(62)

$$\delta_{kt} \delta_{ls} = -\frac{1}{s^2} \underbrace{\left( W^{(Q2)}_{qa} X_{tp} X_{sq} + W^{(K2)}_{pa} X_{tq} X_{sp} \right) X_{ka} X_{lq}}_{g^*_{ktls}}$$

(63)

This correction, even with the stated assumptions, appears highly non-trivial and perhaps intractable in general, even before considering the subsequent values step and multiple stacked heads. Therefore, this generalisation will not be pursued at the current time for attention mechanisms.

Whether this partially explains the absence of normalisation in attention layers is an interesting speculation. In such cases, standard normalisation would not reduce the attention divergence as it does for affine maps; consequently, any empirical advantage may be absent. Perhaps this has implicitly suppressed the use of normalisation here, although the conceptual undesirability of normalisation here and its interactions with softmax may be additional significant reasons for its absence. Future work could elucidate this by examining whether approximate corrections to the divergence are possible or whether typical normalisers help mitigate this, yielding improved performance.

### C.4 Residual Divergence

Residual models directly complicate the approximation assumptions asserted throughout this work. The purpose of this work was to explore whether simple corrections that reduced the affine divergence could improve performance, which, in turn, naturally yielded normalisation-like maps that offered insight into their success.

This hinged on single-layer assumptions to offer computationally straightforward and reasonably unburdensome operations that mitigate the divergence in single layers. Residual networks do not uphold this single-layer picture of interactions; they directly transmit gradients between layers using an identity map, in combination with whatever other operation is present. This directly breaks down the simple single-layer approximation being proposed and significantly complicates the divergence, which itself is strongly dependent on the function that the residual skip bypasses (often nonlinear). Consequently, pursuing residual ideal-effective divergences and structural corrections may be an interesting direction for future work, but it remains out of scope for the current approach.

## D Gradient-Only Approach and Comparison to Natural Gradients

Both the affine divergence mechanism discussed and that of Natural gradients critique the primacy of parameters in optimisation. Yet, they differ from the supplanted optimisation target, intermediate representations, and overall model in several key aspects, and their approaches to resolving such critiques differ accordingly. This section will outline how these could be considered related, then indicate the affine-divergence gradient-only corrections, followed by a more thorough comparison.

When considering gradient-only corrections for the affine divergence in isolation, then stronger parallels are drawn with natural gradients: both act to reweight updates to better tune corrections towards their optimisation target. Except for differing update trajectories, neither structurally changes the model. Therefore, natural gradients could be classified under this paper's gradient-only definition. In this framing, the affine-divergence gradient corrections act as a middle ground approach between standard gradient descent and natural gradients. The gradient-only affine-divergence corrections are as follows.

Similar to structural corrections, one can amend the gradient descent step *only* in various ways. In particular a global-layerwise corrected learning rate, as shown in *Eqn.* 64 can be implemented to keep the update proportions intact.

$$\eta' = \frac{\eta}{\|\vec{x}\|^2 + 1} \tag{64}$$

Or one can apply it locally to just the weight matrices, and an overall halving of layerwise learning rate, as shown in *Eqn.* 65 — producing a disproportionate change in gradients between weights and biases.

$$\eta'_{\mathbf{W}} = \frac{\eta}{2\|\vec{x}\|^2}$$
$$\eta'_{\vec{b}} = \frac{\eta}{2} \tag{65}$$

Hence, the four primary affine divergence solutions form between structural and gradient-only, and each may then be proportional or disproportional.

However, due to the nature of autodiff, accumulating over samples upfront, this method is not implementable unless working with the full Jacobians directly, which is prohibitively expensive. Therefore, although theoretically plausible, in current approaches to backpropagation, it is not implementable, as it would require a sample-wise adjustment to the Jacobian, which is not accessible straightforwardly. A full rebuild of backpropagation using Jacobians could be undertaken in future work, but it is well outside standard practice. This is one reason why structural corrections were predominantly discussed, and exclusively in this work.

Generalising this approach, not in reference to intermediate applications but instead to the final output, then yields natural gradients. This indicates the slight conceptual alignment. However, generally, the approaches differ substantially.

Firstly, as stated, the key difference is which quantity's steepest descent is prioritised with respect to the loss. Classically, these are the parameters, with natural gradients considering the overall model's output, whilst this work prioritises the intermediate activations for steepest descent. An advantage of this middleground is that the steepest descent for activations is available from back-propagation; moreover, it is trivial to derive analytically. However, due to backpropagation, it is not trivially implementable. Natural gradients have been implemented, but they remain computationally expensive.

Secondly, this paper discusses two approaches to solving the affine divergence: gradient-only and structural corrections. This paper investigates the latter for various reasons that fundamentally alter the mathematics of the forward pass to align with ideal-effective representation steps. Whereas, if the natural gradient methodology were to be classified in such a picture, it would *exclusively* constitute the former, gradient-based corrections, rather than structural amendments to the network. Such a structural generalisation of natural gradients does not appear feasible; however, this motivation was never desired by the original works (Amari, 1998) and instead is a framing applied with respect to this paper.

Thirdly, the Reimannian approach also constitutes a distinct consideration from the ideal-effective argument, which follows from propagated parameter corrections as 'effective' changes on subsequent activations. These are two entirely different routes to determine optimality and gradient-only solutions. Divergence corrections and natural gradients do not appear mutually exclusive if their combination is sufficiently motivated.

Fourthly, natural gradient features an invariance to reparameterisations — a property distinctly different from approaches to divergence correction in this work. This is epitomised in the disproportionality discussion of structural corrections, such as RMSNorm's $\sqrt{n}$ factor, which would not change learning trajectories in natural gradients due to reparameterisation invariance.

Hence, although both cases displace the primacy of parameter's steepest-descent updates, natural gradients are not the same as the present method, with orthogonal implementations that differ in their respective priorities, considerations, and approximations, but do share somewhat overlapping conceptual motivation, particularly for gradient-only corrections.

It is argued that the analytical, straightforward 'structural' solutions proposed in this paper represent a more computationally tractable middle ground between approaches; its distinct maps are trivial to implement with slightly altered transforms. This may enable broader applicability.

Moreover, the consideration of primarily structural solutions instead demonstrated that simple adjustments to the forward pass maps, intended to reproduce the mathematically ideal activation update, in turn elucidated a new derivation and mechanistic explanation for normalisers by happenstance. This derivation is only available through the structural approach to the divergence. Hence, explaining the unique mechanistic mode. This alternative explanation also contrasts with prior normalisation literature, since it does not appeal to covariate shift, variance control, or scale-invariance for motivation. Overall, positioning the divergence as a novel framework with potentially significant implications.

## E EXPERIMENTAL DETAILS

A variety of experiments are performed throughout this paper, from fully-connected networks to convolutional architectures. This section details architectural and training methodologies for reproduction and provides justification for several approaches taken.

Beginning with hyperparameters. These were kept as consistent, reasonable values throughout all tests to ensure ablation comparability. Hence, no hyperparameter was optimised collectively or individually, but instead taken as standard values. This was done to ensure consistency, without bias to any particular method — enabling relative comparison rather than absolute performance. Learning rate was chosen to be $\eta = 0.001$ across all runs, with ADAM-optimiser training across 100 epochs for every result, to ensure accuracy stagnation was observed across all results. Batch sizes of $n = 32$ were used unless otherwise stated, and 5 repeats were taken for every result.

The cumulative computation time across all experiments was substantial. Consequently, experiments were performed on CIFAR-10 classification only — selected as a well-studied yet sufficiently challenging benchmark that provides meaningful discrimination between methods. This dataset was not chosen to substantiate the general practicality of these implementations, but instead to assess only the empirical support for the mechanistic claims; hence, CIFAR-10 appeared as a reasonable choice as a standard benchmark. Additional datasets could be considered given greater computational resources, yet the focus of this work is primarily on these theoretical considerations. The central contribution focused on conceptual, with experiments intended to substantiate the proposed mechanisms rather than to provide exhaustive benchmarking for practical applications of the new normalisers. Furthermore, a greater understanding of the stated approximations is encouraged before the corrections can be generalised for widespread applicability — therefore, the results are not to be generalised immediately to current practice.

The formulations for all normalisers and corrections have been provided, or are standard parameterless implementations, whilst the architectures are depicted below in *Fig.* 10.

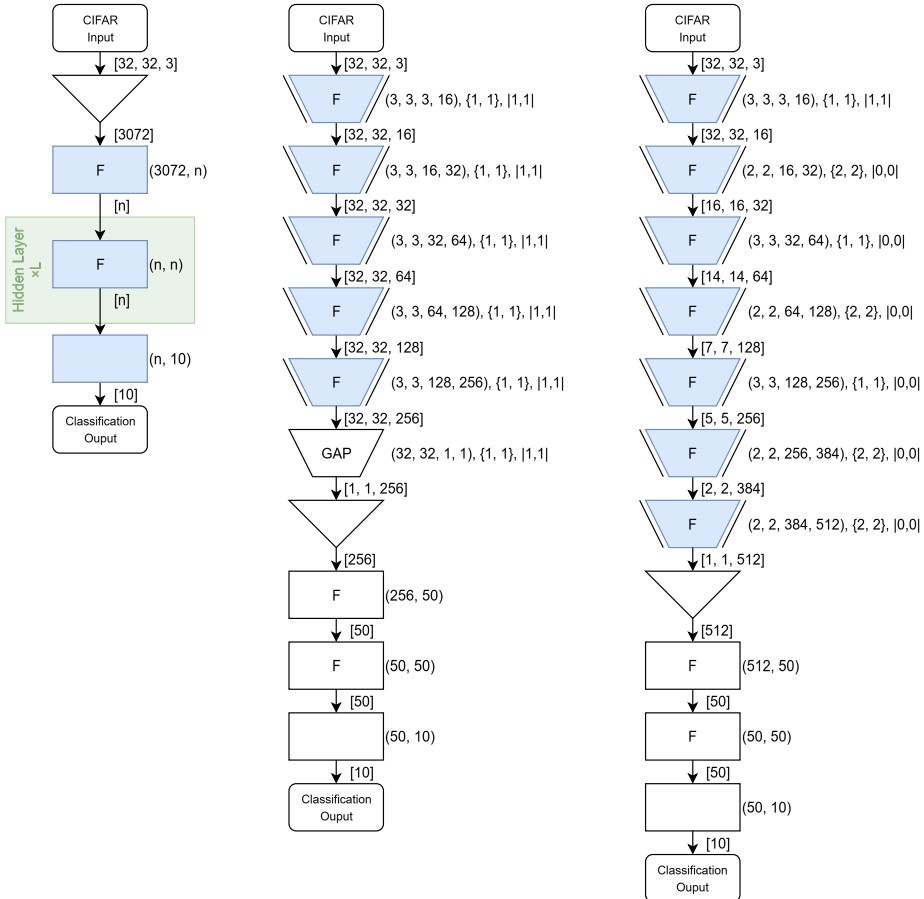

Figure 10: Depicts the three architectures used, displayed using a standardised neural diagrammatic convention (Bird, 2025c). All blue shaded components indicate where the various normalisers or corrections are applied, and $F$ notates the activation function used. These are all precomposed normalisers/corrections for fair comparison with the divergence corrections, which are inseparable or precomposed for affine-like and norm-like, respectively. Leftmost depicts the fully connected architecture, with a green box depicting a hidden layer which may be repeated $L$ times, with layer width $n$. Centre displays the convolutional-GAP architecture used in *Fig.* 8, whilst the rightmost shows the architecture for a sequentially, spatially contracting convolution architecture used in *Fig.* 9.

