# OpenReview forum: "The Affine Divergence: Aligning Activation Updates Beyond Normalisation"
_ICLR.cc/2026/Workshop/GRaM — ICLR 2026 Workshop GRaM Poster_

### Official Review · Reviewer_tDsK · 2026-02-15
**Good paper for this workshop**

**Rating:** 8
**Confidence:** 3

**Review:**

This paper identifies an “affine divergence”: under first-order, single-layer, single-sample analysis, the effective activation update in affine layers scales as (\|x\|^2+1) times the ideal gradient. To align parameter and activation descent directions, it proposes two structural fixes—a norm-like normalization z = W(x/\|x\|) + b and a novel affine-like map z = (Wx+b)/\sqrt{\|x\|^2+1}. This reframes normalization as correcting representation-update misalignment, with CIFAR-10 ablations showing the affine-like map often outperforming parameterless BatchNorm/LayerNorm/RMSNorm.

Pros:

1, The derivation of the affine divergence under explicit single-sample, single-layer, first-order assumptions is mathematically transparent and easy to follow, making the core insight accessible and well-motivated.

2,  By grounding the analysis in geometric considerations and contrasting it with natural gradient methods, the paper offers a conceptually distinct viewpoint on optimization dynamics.

3, By proposing a mechanistic explanation for normalization grounded in activation-update alignment rather than statistical stabilization, the paper offers a fresh theoretical lens on a foundational component of deep learning architectures.

Cons：

1, It would make this work better if the proposed framework were validated on attention-based architectures, such as Transformer models, rather than being limited primarily to fully connected networks.

**Pmlr Suitability:**

Yes

---

### Official Review · Reviewer_KXuG · 2026-02-18
**Review for "The Affine Divergence: Aligning Activation Updates Beyond Normalisation"**

**Rating:** 9
**Confidence:** 5

**Review:**

This paper proposes a new perspective on normalization by analyzing the mismatch between the "ideal" steepest-descent update in activation space and the activation change induced by parameter-space gradient descent. The resulting analysis motivates a normalization-like correction.

Overall, I enjoyed reading this paper. Although I don't get why "the steepest descent in activation space" matters (after all, the "activation space" is a very rough and arbitrary concept, as it's just a middle step in the forward calculation), I do find this work provides a novel, interesting and insightful perspective on the normalization layers in neural networks.

On the technical side, the derivations are detailed and (to the best of my reading) correct. The paper is also well written: the motivation is clear, the key steps are easy to follow, and the exposition does a good job connecting the theory to the proposed constructions.

One small suggestion regarding presentation: it seems you may not need to treat the norm-like and affine-like cases separately. You could simply absorb the bias term $b$ into the weight matrix $W$ by adding an extra dimension to the input $x$ (i.e., define $\tilde{x} = [x; 1]$). This would let you unify the discussion and derivations under a single linear formulation.

**Pmlr Suitability:**

Yes

---

### Meta-Review · Area_Chair_mFyr · 2026-02-26

**Decision:**

Accept

**Metareview:**

This paper studies the geometry of gradient descent updates in activation space. Reviewers find the paper well written, likes the derivations, and appreciate the theoretical results. The topic is also well suited for the workshop.  I recommend acceptance and encourage the authors to incorporate reviewers’ feedbacks, including make the motivation more clear and looking into attention-based architectures.

**Relevance To Proceedings:**

Yes — suitable for PMLR (long paper)

**Relevance To Workshop:**

Yes — suitable for GRaM

---

### Decision · Program_Chairs · 2026-03-02

Accept (Poster)